# When Do Diffusion Models Learn to Generate Multiple Objects?

Yujin Jeong [1]  Arnas Uselis [2]  Iro Laina [3]  Seong Joon Oh [2 4]  Anna Rohrbach [1]

## Abstract

Text-to-image diffusion models achieve impressive visual fidelity, yet they remain unreliable in multi-object generation. Despite extensive empirical evidence of these failures, the underlying causes remain unclear. We begin by asking how much of this limitation arises from the data itself. To disentangle data effects, we consider two regimes across different dataset sizes: (1) concept generalization, where each individual concept is observed during training under potentially imbalanced data distributions, and (2) compositional generalization, where specific combinations of concepts are systematically held out. To study these regimes, we introduce MO-SAIC (**M**ulti-**O**bject **S**patial relations, **A**ttr**I**bution, **C**ounting), a controlled framework for dataset generation. By training diffusion models on MO-SAIC, we find that scene complexity plays a dominant role rather than concept imbalance, and that counting is uniquely difficult to learn in low-data regimes. Moreover, compositional generalization collapses as more concept combinations are held out during training. These findings highlight fundamental limitations of diffusion models and motivate stronger inductive biases and data design for robust multi-object compositional generation. Code and dataset are available at https://github.com/eugene6923/MOSAIC.git.

## 1. Introduction

Diffusion models (Ramesh et al., 2022; Yang et al., 2024a; Chen et al., 2023) have set new standards for visual realism in image generation, yet they remain strikingly unreliable when generating *multiple* objects. While text-to-image diffusion models achieve accuracy above 80% on *single-object* tasks (e.g., generating an object or assigning it a

[1]TU Darmstadt & hessian.AI [2]University of Tübingen [3]Visual Geometry Group, University of Oxford [4]KAIST AI. Correspondence to: Yujin Jeong <yujin.jeong@tu-darmstadt.de>.

*Proceedings of the 43rd International Conference on Machine Learning*, Seoul, South Korea. PMLR 306, 2026. Copyright 2026 by the author(s).

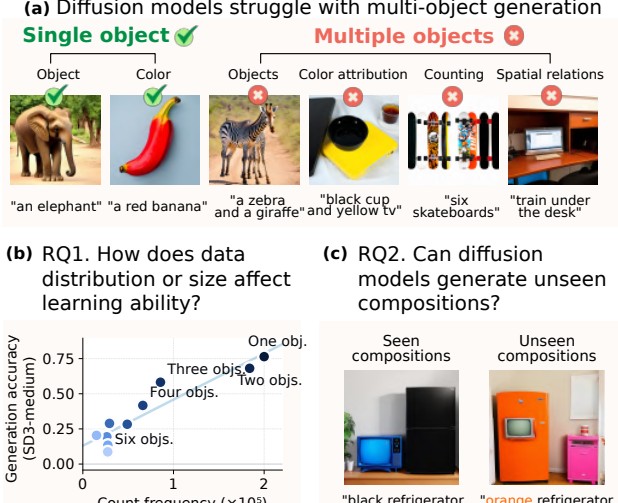

*Figure 1.* **Diffusion models struggle with multi-object compositional generation.** **(a)** Diffusion models generate single object reliably, but struggle with multiple objects. We study two regimes: **(b)** Concept generalization: the model has seen each concept at least once, but may still fail to learn it reliably (e.g., under data imbalance). Generation accuracy is evaluated using Geneval (Ghosh et al., 2023). **(c)** Compositional generalization: the model must generate new combinations of concepts that were never seen together during training. Stable Diffusion 3 (SD3) (Esser et al., 2024) is used to report generation accuracy and qualitative examples.

color), their performance often falls below 50% on *multi-object* tasks in compositional generation benchmarks such as GenEval (Ghosh et al., 2023). As shown in Fig. 1 (a), these shortcomings reveal a critical gap in compositional ability to represent multiple object instances (Counting), correctly bind attributes to each instance (Attribution), and preserve relations between instances (Spatial relations).

Previous works have studied whether diffusion models can generate novel images within the same classes observed during training (Bonnaire et al., 2025; Pham et al., 2025), or exhibit compositional generalization (Okawa et al., 2023; Park et al., 2024; Kang et al., 2024), often in controlled environments. Such studies are valuable for understanding the underlying mechanisms of generalization. However, most of these efforts do not investigate multi-object compositions.

There are multiple possible reasons why multi-object failures may arise in diffusion models, such as, e.g., learning objectives (Wewer et al., 2025; Pogodzinski et al., 2025).

In particular, we focus on the role of training data in models' abilities. We ask the following central question: *How reliably can models generate multi-object compositions under imperfect training data distributions?* While real-world datasets exhibit many intertwined sources of bias and noise, we identify two fundamental data-related failure modes.

The first failure mode concerns data skewness, which may prevent models from reliably learning individual concepts—defined here as the smallest semantic units such as object, color or count. For example, as shown in Fig.1(b), the frequency of <count> in LAION-2B captions (Schuhmann et al., 2022), where a <count> is filtered to represent the quantity of objects, correlates with the counting generation accuracy of Stable Diffusion3 (Esser et al., 2024) (SD3), suggesting that limited exposure to certain counts may impact performance (see Appendix A.1.1 for details).

Based on this observation, we formulate our first research question. *(RQ1) Concept generalization*: The model has seen each relevant concept at least once during training; can it reliably learn these concepts? How does data (concept) imbalance affect learning ability?

The second failure mode concerns whether a model can correctly recombine known concepts when specific compositions are absent during training (Fig. 1c). Although recent advances in dense and accurate captions can improve vision-language alignment (Kim et al., 2023; Elmaaroufi et al., 2025), real-world captions inherently cover only a limited fraction of all possible concept combinations during training. Understanding how unseen compositions affect model performance is therefore important; however, in real-world datasets, it is challenging to accurately assess whether concepts appear in specific compositions.

We therefore pose *(RQ2) Compositional (Combinatorial) Generalization*: Given that all individual concepts are sufficiently observed, can the model recombine known concepts into unseen compositions? How does this ability change as more compositions are held out during training?

Beyond data skewness and held-out compositions, dataset size itself plays an important role. While more data is typically beneficial, even at the scale of billions of samples, certain concepts may still appear rarely in absolute terms (Deitke et al., 2025). Still, dataset size fundamentally shapes both concept learning and compositional generalization, motivating an analysis across multiple data scales.

To systematically analyze the causal effects of dataset properties, we construct a diagnostic dataset, MOSAIC (**M**ulti-**O**bject **S**patial relations, **A**ttr**I**bution, **C**ounting), which explicitly parameterizes object counts, color attribution, and spatial relations as separate factors. Using MOSAIC, we train two diffusion architectures representing earlier and more recent diffusioin variants, without introducing additional inductive biases (e.g., layout conditioning), in order to identify when such biases become necessary. Our findings are summarized as follows:

- **Concept generalization.** Concepts reliably generalize in multi-object scenarios once the dataset is sufficiently large. In low-data regimes, however, increasing scene complexity degrades performance more strongly than concept imbalance, especially for Counting.
- **Compositional generalization.** Diffusion models increasingly fail to exhibit compositional generalization as more concept combinations are held out during training. The difficulty of recombining concepts compositionally follows an ordering: Attribution < Counting < Spatial Relations.
- **Generalization of findings to more realistic visual settings.** Our findings extend to visually richer data with object appearances and occlusions. Specifically, (i) counting remains brittle when fine-tuning SD3, and (ii) compositional generalization continues to degrade as more compositions are not observed during training under object co-occurrence scenarios.

## 2. Related Work

**Multi-object failures in diffusion models.** Recent text-to-image diffusion models (Ramesh et al., 2022; Yang et al., 2024a; Chen et al., 2023) have demonstrated impressive performance in generating realistic images. However, benchmarks (Huang et al., 2023; Ghosh et al., 2023; Jeong & Uselis et al., 2025), which evaluate models whether generated images satisfy compositional constraints given a text prompt, confirm that foundational diffusion models (Rombach et al., 2022; Podell et al., 2023; Esser et al., 2024; Xiao et al., 2024) consistently fail in multi-object settings. This has motivated methods (Kang et al., 2025b; Binyamin et al., 2025; Boo et al., 2025; Yoo et al., 2025; Han et al., 2025; Chefer et al., 2023; Chen et al., 2024) that build on top of these foundational models to mitigate the issue through attention guidance or layout control, rather than analyzing their underlying causes. Some works attribute these failures to frequency-related effects in training data (Malakouti & Kovashka; Kang et al., 2025a) or limitations of text encoders (Toker et al., 2024; Tong et al., 2023), but they do not systematically control the training data. In contrast, we study these failures under controlled multi-object training distributions, enabling causal analysis of data effects.

**Compositional generalization in image diffusion models.** Compositional generalization has been widely studied in discriminative models (Uselis et al., 2025; Thrush et al., 2022; Wiedemer et al., 2025; Ma et al., 2023; Li et al., 2023). On the generative diffusion side, most prior work focuses on in-distribution (ID) generalization, evaluating whether models can generate novel images within the training distribution

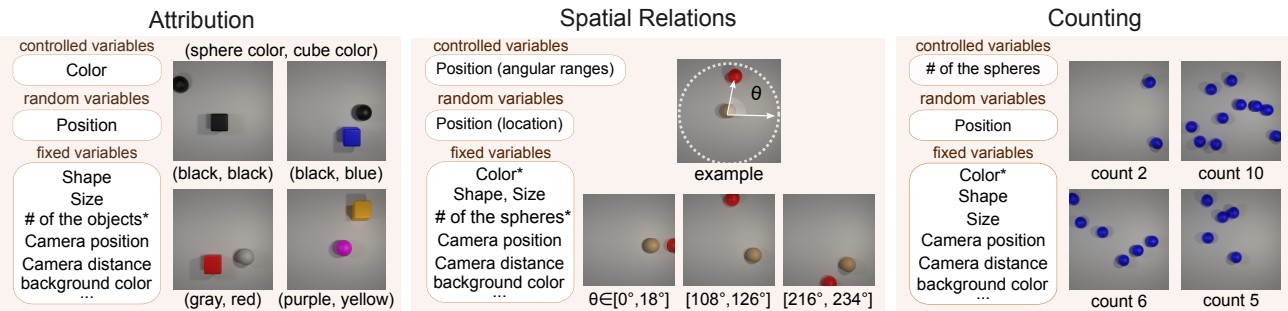

Figure 2. **Our controlled dataset MOSAIC is designed for analyzing multi-object compositions.** Each subset isolates a specific reasoning dimension by varying one factor while randomizing others. (i) Attribution: varies object colors while keeping positions randomized, enabling control over color–object associations (e.g., "black sphere and blue cube"). (ii) Spatial Relations: varies the angular placement between two objects while fixing color and shape. (iii) Counting: varies the number of spheres while keeping all other factors constant. We show the BASE settings; an asterisk (*) marks variables that can be varied in other dataset variants.

rather than testing true compositional generalization (Bonnaire et al., 2025; Pham et al., 2025; Garnier-Brun et al., 2025; Kamb & Ganguli, 2024). Only a few studies explicitly investigate compositional generation by carefully controlling the training data (Okawa et al., 2023; Park et al., 2024; Yang et al., 2024b; Farid et al., 2025), and report emerging compositional generalization in diffusion models. However, these works primarily evaluate single-object settings with continuous inputs (e.g., RGB values), which yield a large space of possible compositions between concepts and do not reflect the discrete, multi-object compositional challenges. More recently, Bradley et al. (Bradley, 2025) examine object length generalization, but their model relies on explicit spatial conditioning, making the setting less comparable to unconstrained real-world generation. In contrast, we focus on vanilla diffusion models, and analyze when such inductive biases (e.g., spatial priors) become necessary.

**Controlled compositional datasets.** Several controlled compositional datasets, such as Shapes2D (Okawa et al., 2023), 3D Shapes (Burgess & Kim, 2018), and CelebA (Liu et al., 2015), focus on single-object scenes. Other datasets based on CLEVR (Johnson et al., 2017), such as Kubric (Greff et al., 2022), Super-Clevr (Li et al., 2023), and CLEVR-X (Salewski et al., 2020), provide rich multi-object scene annotations (e.g., object locations, segmentation masks, language explanations, and depth), but do not explicitly factorize multi-object compositional concepts. More recently, COMFORT (Zhang et al., 2024) introduces an evaluation protocol to study spatial language understanding with a simulator. Our dataset, MOSAIC, is built on top of COMFORT, and is designed to disentangle multi-object compositions.

## 3. MOSAIC: Diagnostic Dataset for Multi-Object Compositions

MOSAIC is the first controlled dataset that isolates three specific multi-object compositional concepts: (Color) Attribution, Counting, and Spatial Relations.

### 3.1. Default Design

Figure 2 shows how the dataset is constructed. Each subset varies one factor (e.g., object color, relative position, or number of instances) while keeping other properties (e.g., lighting, camera viewpoint) fixed or randomized. To ensure a simplified and controlled setting, we avoid occlusions between objects. MOSAIC is generated using a simulation-based pipeline (Zhang et al., 2024), which uses Blender to render photorealistic scenes with explicitly controlled parameters and exact ground-truth annotations. Details are in Appendix A.1.2. We begin by describing the Base setting.

**Attribution (Figure 2, left).** This subset isolates *attribute binding* between object type and color by fixing the object identities (e.g., "black sphere and red cube"). We use two fixed object identities, *sphere* and *cube*, so that attribute binding is directly evaluated between object type and color. We use ten distinct colors, yielding $10 \times 10 = 100$ possible sphere-cube color combinations.

**Spatial Relations (Figure 2, middle).** This subset isolates *relative spatial* layouts between two objects. A "brown" reference sphere is fixed at a random position, and a second sphere is placed on the same horizontal plane at one of ten angular intervals around the "brown" sphere. Specifically, we discretize the full circle into ten 18° ranges (measured counterclockwise starting from the 3 o'clock direction) with 18° gap between the intervals, yielding ten spatial relation classes. (e.g., "the red sphere is at 216° relative to the brown sphere"). The angular relation is fixed per class, while the distance between objects is randomly decided.

**Counting (Figure 2, right).** This subset isolates object numerosity by varying the number of object instances from one to ten while keeping other visual factors fixed (e.g., "ten spheres"). Here, higher counts introduce greater spatial complexity, requiring the model to maintain clear separation between repeated objects rather than collapsing them into fewer instances.

## 3.2. Dataset Variants

MOSAIC provides flexible control over several factors, allowing us to systematically vary dataset difficulty. Starting from the `Base` setting, we introduce variants of `Complex` and `Grid` that adjust scene complexity across different tasks. We further introduce a `Composition` setting to evaluate compositional generalization with multiple conditioning concepts.

**Scene complexity scaling.** We introduce the `Complex` setting for Attribution and Spatial Relations to systematically scale scene complexity. This allows us to analyze how scene complexity affects learning, as Attribution and Spatial Relations involve only two objects, whereas Counting includes a wider range of object counts (from one to ten). Scene complexity increases as the number of objects in the scene varies. For Attribution, we increase the number of objects by duplicating the existing object categories (i.e., spheres and cubes), while preserving their attributes. The total number of objects is constrained to range between 2 and 10. For Spatial Relations, we introduce additional objects (e.g., blue spheres) as distractors, with the total number of objects randomly varied up to 10. Both `Complex` settings match the maximum scene complexity of the Counting `Base` task.

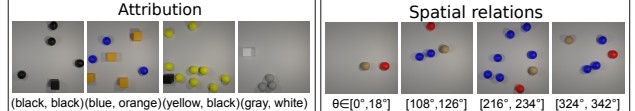

*Figure 3.* **`Complex` settings for Attribution and Spatial Relations.** Scene complexity is increased by introducing additional objects: for Attribution, objects are duplicated, while for Spatial Relations, additional objects are added as distractors.

**Spatial grid layout variant.** We additionally introduce the `Grid` setting, which reduces scene complexity for the Counting task. Although Counting inherently involves a larger number of objects than the `Base` settings for Attribution and Spatial Relations, we aim to control the increasing degrees of freedom as object count grows. To this end, we impose a radial grid layout that constrains object positions to predefined regions, thereby introducing an explicit spatial prior. This reduces positional variability and simplifies the task. While the grid layout can be applied to all tasks, as it restricts objects to limited spatial regions, we focus on the Counting task in Fig. 5. In this setting, each object is placed within a designated radial cell with small positional jitter.

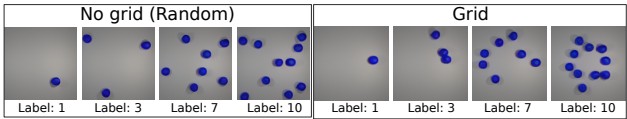

*Figure 4.* **`Grid` setting for Counting.** Objects are constrained to predefined radial cells with small positional jitter, reducing positional variability compared to the default setting where objects can appear anywhere in the image.

*Table 1.* **Summary of design choices for concept and compositional generalization experiments.** We list the dataset sizes, distribution types, and conditioning factors used for Attribution, Spatial relations, and Counting in each setting.

| **Concept generalization** | Attribution | Spatial relations | Counting |
|---|---|---|---|
| Dataset size / Distribution | 2k, 10k, 20k, 100k / Uniform or Skewed | | |
| Condition | 10 sphere colors, 10 cube colors | 10 relations | 10 counts |
| Evaluation | Accuracy (+ Memorization rate) | | |
| Experimental settings | `Base`, `Complex` | `Base`, `Complex` | `Base`, `Grid` |
| **Compositional generalization** | Attribution | Spatial relations | Counting |
| Dataset size / Distribution | 10k, 20k, 100k / Uniform | | |
| Condition | 10 sphere colors, 10 cube colors | 10 sphere colors, 10 relations | 10 sphere colors, 10 counts |
| Evaluation | Attribution accuracy | Joint accuracy (Color & Relation acc) | Joint accuracy (Color & Count acc) |
| Experimental settings | `Base` | `Comp` | `Comp` |

**Compositional generalization setting.** In addition, to evaluate compositional generalization, we require at least two conditioning concepts; we therefore introduce the `Composition` setting. Attribution already provides two independent concepts (sphere color × cube color) in `Base` settings, whereas Counting and Spatial Relations originally involve only a single conditioning factor (count or angle). Therefore, we introduce `Composition` settings for Counting and Spatial Relations. Specifically, we introduce an additional *Color* factor to both tasks (marked with an asterisk (*) in Fig. 2), forming conditioning pairs: (color × count) and (color × spatial relation). Using color as a shared factor ensures comparable task difficulty across settings, while enabling us to analyze whether diffusion models exhibit preferences for certain cues when recombining concepts. We use the same set of ten colors as in Attribution to maintain consistency across tasks. For Counting, all objects share the same color, sampled from this set. For Spatial Relations, one reference sphere remains "brown", while the second sphere varies in color. More examples are provided in Appendix Fig.A.27.

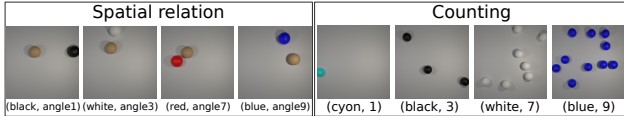

*Figure 5.* **`Composition` setting for Spatial relation and Counting.** We add Color as an additional conditioning factor, forming compositional pairs of color × spatial relation and color × count.

## 4. Experimental Setup

In this section, we first describe the experimental designs (Sec. 4.1) for our two research questions, and then describe our training and evaluation setup (Sec. 4.2).

### 4.1. Experimental Designs

We define two key experimental design choices. The first corresponds to concept generalization (RQ1) and the sec-

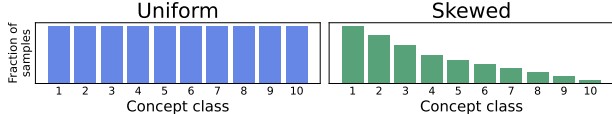

*Figure 6.* **Example of concept imbalance settings.** *Uniform*: all categories have the same number of samples. *Skewed*: the frequency of categories varies, with some categories appearing more often than others.

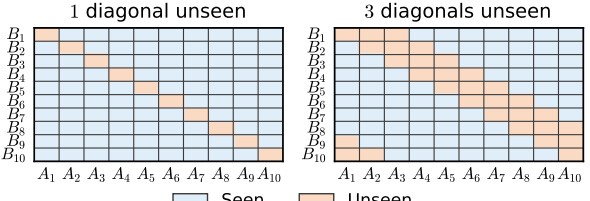

*Figure 7.* **Seen and unseen composition configurations**. Each matrix enumerates all possible combinations between two concepts $A_i$ and $B_j$ (e.g., Color × Count). Cells in blue indicate combinations that are observed during training, while cells in orange indicate unseen (held-out) compositions. We vary the number of diagonals removed, which controls how many concept pairs are never seen during training. This allows us to evaluate whether diffusion models can generalize to unseen concept compositions even when all individual concepts are fully observed.

ond corresponds to compositional generalization (RQ2). We vary dataset size, concept (im)balance, and the number of unseen compositions independently under each setting, enabling a comprehensive controlled analysis. The overview is in Table 1 and further details are provided in Appendix A.1.2.

**Concept imbalance.** To investigate RQ1, we construct two concept imbalance regimes: *skewed* and *uniform* (see Fig.6). The *skewed* distribution is inspired by the frequency patterns observed in LAION-2B(Schuhmann et al., 2022), as illustrated in Fig. 1b. While such imbalance patterns are naturally defined for counts, we extend the same structure to other factors (e.g., angles and colors) to enable consistent and controlled comparisons across tasks. This allows us to systematically study how data imbalance affects model performance beyond counting. Evidence of similar biases in spatial relations is further illustrated in Appendix Sec. A.1.1.

Each distribution is scaled proportionally to preserve its relative (im)balance pattern. In contrast, the *uniform* setting assigns an equal number of samples to each category. Imbalance is applied at the task-relevant categorical level: counts for Counting (lower counts are more frequent in *skewed*), angle intervals for Spatial Relations (smaller angles are more frequent), and color-pair combinations for Attribution (imbalance is applied to sphere colors while cube colors remain uniform). We use the following ordered color set: RED, GREEN, BLUE, YELLOW, PURPLE, ORANGE, CYAN, GRAY, WHITE, BLACK.

We vary the dataset size across 2k, 10k, 50k, and 100k samples, following prior work on in-distribution generalization and memorization (Pham et al., 2025), where performance typically saturates around 100k examples.

**Seen vs unseen compositions.** Under RQ2, we evaluate compositional generalization by combining two seen concepts, inspired by (Uselis et al., 2025). Each individual concept (e.g., color "red" or count "2") is fully observed, but specific pairs are held out so the model must recombine known concepts at test time. As shown in Fig. 7, we vary the compositional difficulty by holding out different numbers of diagonals (0, 1, 3, 5, or 8) from the matrix.

We train with dataset sizes of 10k, 50k, and 100k, keeping a uniform distribution over seen pairs to avoid confounding effects from frequency imbalance among compositions.

Since removing diagonals changes the number of available compositions, we resample the remaining ones to keep the total dataset size comparable.

### 4.2. Training and Evaluation Setup

An overview of our training and evaluation pipeline is illustrated in Figure 8.

**Training.** We use a latent diffusion model consisting of a pretrained VAE (Kingma & Welling, 2013), a diffusion backbone, either a U-Net (Ronneberger et al., 2015) or a Diffusion Transformer (DiT) (Peebles & Xie, 2023), and a lightweight condition encoder. The diffusion backbone follows the small latent diffusion architecture (Rombach et al., 2022) (approximately 90M parameters), where conditioning is injected exclusively through attention layers. For training, the U-Net is optimized using a score-matching objective (Song et al., 2020), while DiT is trained with a flow-matching objective (Lipman et al., 2023), mirroring the training objectives used in Stable Diffusion 2.0 (Rombach et al., 2022) and 3-m (Esser et al., 2024), respectively. We adopt the pretrained Stable Diffusion 2.0 VAE (Kingma & Welling, 2013) to encode images into latents, keeping the VAE frozen during training. We additionally verify qualitatively that the VAE preserves conditioning accuracy on MO-SAIC. Each conditioning variable (e.g., count) is represented as a one-hot vector and encoded with a small multi-layer encoder. If the concept has two classes (e.g., attribution), we encode each one independently and concatenate them as two tokens before passing them to the diffusion attention layers. We also explored text-embedding conditioning, which is described in Appendix A.2.3. The diffusion models and the condition encoder are trained jointly. We report results from the checkpoint that achieves the best validation accuracy. Training details are in Appendix A.1.3.

**Evaluation.** To assess if the model can correctly generate multi-object scenes, we train task-specific discriminative classifiers. For Counting, we train a discriminative CNN

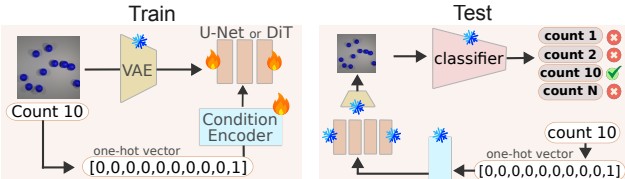

*Figure 8.* **Training and evaluation pipeline.** During training (left), a one-hot condition vector (e.g., "count = 10") is embedded by a condition encoder and integrated into the diffusion model, such as a U-Net or a Diffusion Transformer (DiT), via attention with the VAE-encoded latent representation. The condition encoder and the diffusion model are trained jointly. During evaluation (right), the trained diffusion model generates samples from a target condition vector, and a pretrained classifier determines whether each output matches the intended condition (Correct / Incorrect).

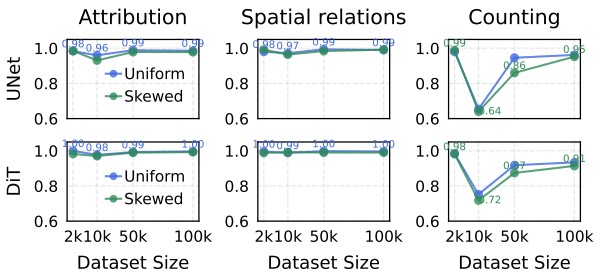

*Figure 9.* **Accuracy vs. dataset size and data distribution.** Attribution and Spatial Relations achieve high accuracy across data regimes, but Counting fails at 10k and 50k data scales, neither memorizing nor generalizing especially under skewed distributions, and recovers when the dataset is large.

classifier (O'shea & Nash, 2015) from scratch. For Attribute Binding and Spatial Relations, we use ResNet-based (He et al., 2016) classifiers initialized from ImageNet pretraining. All classifiers are trained using cross-entropy loss with standard data augmentation. We verify that classification accuracy on VAE-reconstructed images remains near 100%, as reported in Appendix Table A.6. For RQ1, we primarily evaluate accuracy using a trained classifier and further analyze memorization rate using pixel-wise distances between training and generated images. For RQ2, we additionally measure joint accuracy for Counting and Spatial Relations by training an extra 10-class color classifier. Further implementation details are provided in Appendix A.1.4.

## 5. Concept Generalization

We first address RQ1 *(Concept Generalization)*: whether a model can correctly generate a concept that has been observed at least once during training. In real-world datasets, data distributions are often imbalanced (e.g., skewed) rather than uniform; we refer to this as concept imbalance (Figure 6). We ask how dataset size and concept imbalance affect a model's ability to generate the concepts.

**Concept generalization can emerge with sufficient data**

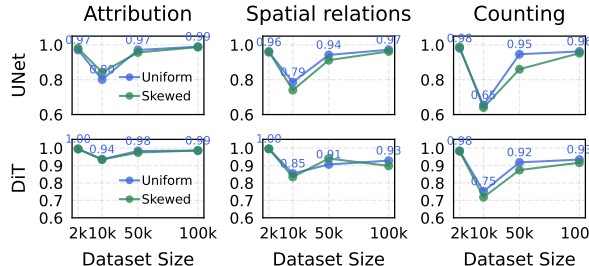

*Figure 10.* **Scene complexity becomes critical in low-data regimes.** We increase the number of objects for Attribution and introduce additional blue spheres as distractors for Spatial Relations. (Bottom) At a dataset size of 10k, accuracy drops for both Attribution and Spatial Relations, although the degradation is less severe than for Counting.

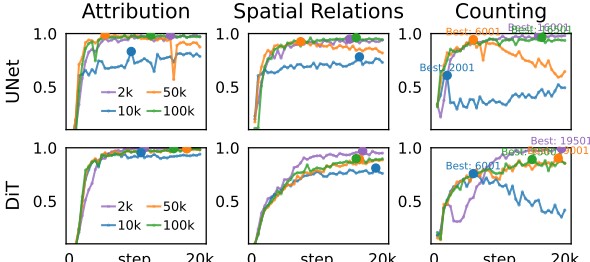

*Figure 11.* **Accuracy trajectories across training steps.** Attribution and Spatial Relations use the `Complex` setting, while Counting uses the `Base` setting. (Top) With a U-Net backbone, Counting exhibits early peaking and subsequent degradation at dataset sizes of 10k and 50k, while Attribution and Spatial Relations remain stable. (Bottom) With a DiT backbone, a similar early-peaking behavior is observed for Counting at 10k, whereas other tasks remain stable.

**size regardless of concept imbalance.** We begin with the simple `Base` settings to assess whether models can reliably learn each concept. Figure 9 shows clear differences among Attribution, Spatial Relations, and Counting. Attribution and Spatial Relations remain stable across all dataset sizes and data distributions, achieving over 90% accuracy even under highly skewed training distributions across both architectures (U-Net and DiT). In contrast, Counting exhibits a distinct behavior. At the smallest dataset size (2k), the models reach near-perfect accuracy, but then performance drops sharply at intermediate scales (10k and 50k), and only gradually recovers as the dataset size increases further (100k), largely independent of data skewness. This trend is further analyzed using the *memorization rate* (Appendix Fig. A.3). At small dataset sizes, all tasks exhibit near-complete memorization that decreases as scale increases; however, Counting shows a transition regime where memorization becomes infeasible while generalization has not yet emerged.

**Scene complexity plays a dominant role.** Counting differs

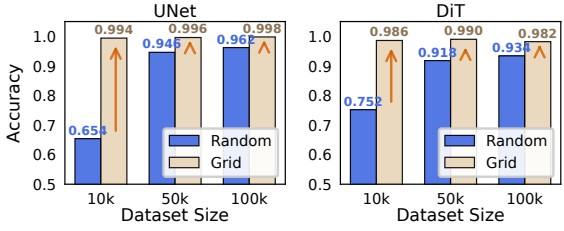

*Figure 12.* **Introducing a spatial prior stabilizes counting performance.** Using a grid layout dramatically improves counting accuracy across all dataset sizes and distributions.

from Attribution and Spatial Relations in terms of scene complexity, as it involves a larger number of objects (up to 10), whereas the others contain only two objects. This raises the question of whether Counting is inherently more difficult, or whether performance degradation is primarily driven by increased scene complexity. To disentangle these factors, we train the model with the `Complex` setting, which systematically increases scene complexity for Attribution and Spatial Relations (Section 3.2). For a dataset size of 10k (Figure 10), now accuracy also drops for Attribution and Spatial Relations, although the degradation remains less severe than for Counting. Yet this confirms that increased scene complexity substantially contributes to performance degradation in low-data regimes. We additionally observe that DiT, a more recent architecture trained with modern learning objectives, exhibits improved robustness compared to U-Net under limited data.

**Counting exhibits distinct learning dynamics.** To investigate why Counting degrades more severely in low-data regimes compared to other concepts, we analyze how accuracy evolves during training (Fig. 11). For a fair comparison, we use the `Complex` settings for Attribution and Spatial Relations. While Attribution and Spatial Relations quickly saturate, Counting peaks early and subsequently deteriorates. In contrast, the training loss decreases smoothly across all dataset sizes and architectures (Appendix Fig. A.5). We further analyze this trend using per-class accuracy (Appendix Fig. A.6), which indicates that performance on higher object counts collapses first. This discrepancy suggests a misalignment between optimization and task performance: the model continues to minimize the training objective while progressively losing its ability to maintain multiple distinct objects when data size is not sufficient.

**Lowering spatial complexity facilitates counting.** To further probe the factors underlying counting performance at moderate dataset sizes, we simplify the task by introducing a radial grid layout to the training dataset, as introduced in Sec. 3.2, where each object is constrained to appear within a designated region of the canvas (with a small positional jitter). Under this reduced spatial complexity, counting accuracy increases substantially across all dataset sizes and data distributions (Figure 12). This suggests that imposing simplified spatial structure can greatly facilitate accurate

counting in generation, especially in low-data regimes, highlighting the importance of inductive bias for robust performance.

> **Takeaway 1:** Concept learning exhibits different behaviors depending on scene complexity and dataset size, rather than concept imbalance. Under limited data regimes, Counting accuracy peaks early and subsequently deteriorates, whereas Attribution and Spatial Relations converge to suboptimal performance without such instability. The difficulty of learning Counting can be partially alleviated by reducing scene complexity (e.g., fewer objects or simpler spatial layouts).

## 6. Compositional Generalization

In this section, we investigate RQ2: *Compositional generalization*. Prior diffusion studies report successful compositional generalization in single-object settings with continuous attributes (Okawa et al., 2023; Park et al., 2024; Yang et al., 2024b), which implicitly provide dense coverage of attribute combinations that is not representative of real-world settings. In contrast, we explicitly control the number of unseen compositions (Figure 7) and dataset size. We focus on the DiT architecture, with corresponding U-Net results provided in the Appendix Sec. A.2.2. We use the `Base` setting for Attribution and the `Composition` settings for Spatial Relations and Counting.

**Performance degrades as unseen compositions increase, with limited gains from data scaling.** Figure 13 varies the number of held-out (unseen) compositions (x-axis) and the training set size (10k / 50k / 100k) across columns, using the diagonal leave-out scheme. Figure 13 (top) reports accuracy on *seen* compositions. Attribution and Spatial Relations remain consistently strong across all dataset sizes. Counting, however, exhibits lower and more variable performance across all dataset sizes, consistent with the trends observed in Section 5. Note that we introduce additional color conditions in this setting; therefore, the absolute performance is not directly comparable to the results in Section 5.

Figure 13 (bottom) reports accuracy on *unseen* compositions, which directly probes compositional generalization. Across all tasks, accuracy improves with increasing dataset size. As more combinations are held out (along the x-axis), performance declines across all categories and dataset sizes, which is consistent with prior observations in discriminative models (Uselis et al., 2025). The degree of degradation differs across concepts; while Attribution remains the most robust, Spatial relations degrade most sharply.

**Spatial relations are more difficult to learn compositionally.** To better understand this failure mode, we analyze generated images and confusion matrices when half of the

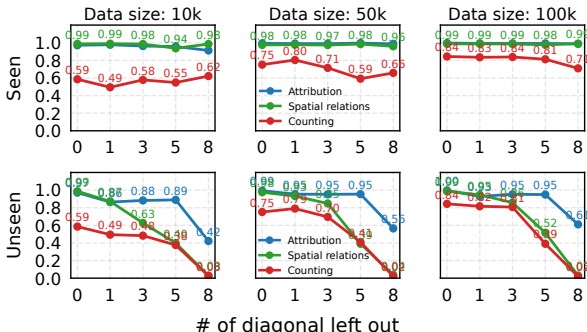

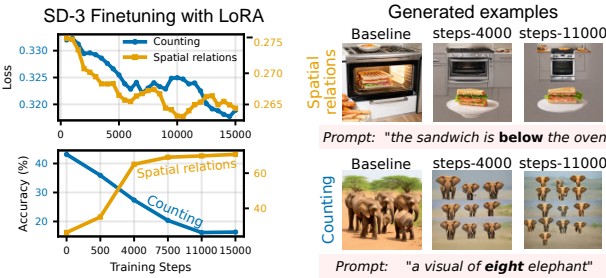

Figure 15. **Fine-tuning behavior of SD3-medium on SPEC for spatial relations and counting.** (Left) Training dynamics for each subset, where the top row shows training loss and the bottom row shows evaluation accuracy. (Right) Qualitative generation examples: the top row shows spatial relation samples, and the bottom row shows counting samples. While training loss consistently decreases for both tasks, counting accuracy degrades, whereas relative spatial accuracy increases, a trend that is also reflected in the qualitative generation examples.

Figure 13. **Compositional generalization on dataset size and the number of unseen compositions.** (Top) For seen compositions, Attribution and Spatial relations remain stable across all dataset sizes, while Counting improves noticeably as the dataset size increases. (Bottom) For unseen compositions, performance drops rapidly as the dataset size decreases or the number of held-out compositions increases.

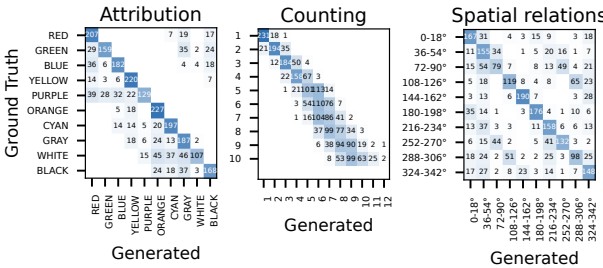

Figure 14. **Confusion matrix on unseen diagonals when half of the compositions (5 diagonals) are unseen.** Compared to Attribution and Counting, Spatial relations show no clear error pattern.

compositions are held out (Figure 14), and we focus subsequent analyses on the 100k dataset unless otherwise stated. For Attribution, the model exhibits highly localized confusion patterns aligned with perceptual color similarity (e.g., purple shifts toward red and blue). For Counting, the model typically predicts one more or fewer instances than the target count. In both tasks, predictions remain "near" the correct label, indicating that the underlying concepts are at least partially learned. In contrast, Spatial Relations display broad and less structured confusion across angular bins, revealing substantial difficulty in learning disentangled geometric relations. Overall, these results suggest a consistent hierarchy in the difficulty of compositional concept recombination: Attribution < Counting < Spatial Relations.

**Takeaway 2:** As the number of held-out compositions increases during training, compositional generalization degrades, especially for Spatial Relations. This suggests that compositional generalization reflects a fundamental limitation of current diffusion models.

## 7. Generalization to More Realistic Settings

In the previous section, we investigated two research questions: (RQ1) How does concept imbalance affect a model's ability to learn individual concepts? (RQ2) How does compositional generalization degrade as the number of unseen concept combinations increases? Based on these questions, we identified two key findings in our MOSAIC experiments: (i) counting was difficult to learn under limited data regimes rather than due to concept imbalance in Sec. 5, and (ii) compositional generalization degraded as more concept combinations were held out during training in Sec. 6. Since MOSAIC was intentionally designed as a highly controlled diagnostic benchmark, we next evaluate whether these trends persist under more realistic settings. Additional results, including experiments with text-prompt conditioning and more complex object/background variations (e.g., cars), are provided in Appendix Sec. A.2.3.

**Fine-tuning behavior for concept generalization: Counting vs. Spatial Relations.** Our controlled experiments in Section 5 are based on training from scratch. To extend these findings to a more realistic setting, we study fine-tuning behavior using pretrained diffusion models such as SD3-medium (Esser et al., 2024). Training large-scale models from scratch is impractical; therefore, we analyze fine-tuning dynamics using LoRA (Hu et al., 2021).

To examine whether the observed dynamics persist, we conduct additional experiments on the SPEC benchmark (Peng et al., 2024). SPEC is designed for text–image retrieval tasks involving *relative spatial relations* and *counting*, and thus provides aligned image–text pairs. The images in SPEC contain visually richer scenes with realistic backgrounds, object appearances, and greater intra-scene variability compared to MOSAIC (examples in Appendix Fig. A.29). From SPEC, we construct datasets for Counting and Spatial Relations and extract 1.5K image–text pairs for training and to maintain

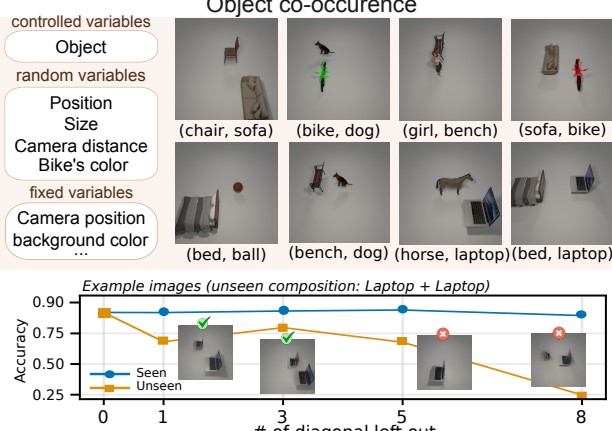

*Figure 16.* **Compositional generalization under realistic object co-occurrence settings.** (Top) Example scenes from the less controlled MOSAIC OBJECTS variant, with object co-occurrence. (Bottom) Accuracy on seen and unseen object compositions as the number of held-out diagonal compositions. While performance remains high on seen compositions, accuracy on unseen compositions degrades rapidly as more combinations are held out. An illustrative example is the unseen composition (laptop, laptop).

consistency with our RQ1 setup, we use the same prompts for generation to test the generation accuracy. Generation accuracy is evaluated using the Geneval framework (Ghosh et al., 2023), which leverages detection models to assess counting and spatial relation accuracy (Fig. 15, left). We verify that the object categories in SPEC are compatible with those used to train the detection models.

Figure15 (left) shows the training loss (top) and evaluation accuracy (bottom) over training steps, where blue curves correspond to Counting and yellow curves to Spatial Relations. While the training loss decreases for both tasks, spatial relation accuracy continues to improve, whereas counting accuracy deteriorates on evaluation prompts. Qualitative examples (Fig. 15, right) further illustrate this behavior, comparing the baseline (no fine-tuning) with models at 4k and 11k training steps. These results indicate that spatial relations benefit steadily from fine-tuning, while counting remains unstable and often incorrect. The observed trends are consistent across hyperparameter variations (Appendix Fig. A.15). While this experimental settings are a bit different from our controlled settings (e.g., frozen conditioning encoders, tuning with LoRA and pretrained dataset scale) may vary in practice, our goal is to understand how performance differs under more realistic training scenarios. Overall, these findings align with our results on MOSAIC, reinforcing that counting remains challenging even under more diverse data and realistic training settings.

**Compositional generalization under more diverse settings: Object co-occurrence.** In the simplified setting without scene complexity, compositional generalization remains highly challenging. We therefore extend our evaluation to a more diverse and less constrained setting, where scenes contain varied object categories with different sizes and appearances. Since publicly available benchmarks are typically limited to a fixed set of detectable object categories, we construct a less controlled variant of MOSAIC based on the Comfort-Car (Zhang et al., 2024) dataset. Compared to the original setting, this variant introduces realistic object shapes, varying camera distances (inducing depth and scale changes), and frequent inter-object occlusions, resulting in a distribution shift in appearance, viewpoint, and occlusions.

In this setup, the model is required to generate pairs of object categories that never co-occur during training. We select 10 object categories and place two objects per scene, following the same compositional protocol as in Figure 7. The task formulation remains simple, but the setting introduces greater diversity and reduced control compared to our earlier experiments.

Representative examples are shown in Figure 16 (top). Figure 16 (bottom) reports accuracy as the number of held-out compositions increases. The classifiers are trained to evaluate if the two objects are available. Consistent with our controlled experiments, performance degrades substantially on unseen object pairs. Qualitatively, the model often collapses to generating a single instance or produces incorrect secondary objects. Overall, this demonstrates that the compositional generalization gap observed in controlled settings persists in less controlled scenarios.

> **Takeaway 3:** The behaviors observed in controlled settings persist under more diverse and realistic conditions, including variations in appearance, viewpoint, and occlusion: (1) Counting remains unstable when fine-tuning Stable Diffusion 3, and (2) Compositional generalization under object co-occurrence settings continues to degrade as the number of unseen object compositions increases, often collapsing to a single object.

## 8. Conclusion

We investigated how data properties contribute to the limitations of multi-object generation across Spatial Relations, Counting, and Attribution. Using our MOSAIC dataset generation framework, we studied both concept generalization (RQ1) and compositional generalization (RQ2). For RQ1, we found that all tasks eventually generalize at a sufficient scale. However, Counting is particularly fragile in low-data regimes, and reducing scene complexity can only partially mitigate this issue. For RQ2, compositional generalization collapses as the number of unseen combinations increases, with Spatial Relations being particularly affected. Overall, our findings indicate that current diffusion models lack mechanisms for multi-object compositional generation.

## Impact Statement

This work aims to advance the scientific understanding of how data properties influence compositional generalization in conditional diffusion models. By providing systematic diagnostics and controlled benchmarks, our findings can support the development of more reliable generative models, particularly for multi-object generation tasks where robustness and interpretability remain limited. These insights may benefit downstream applications that rely on controllable image synthesis, such as data generation for simulation, education, and content creation, by improving model reliability and reducing unintended generation failures. As with most generative modeling research, there exists a potential risk that improved generative capabilities could be misused to create misleading or fabricated visual content. However, our work focuses on diagnostic analysis rather than deploying or releasing high-fidelity real-world generation systems. Our experiments are conducted on controlled or synthetic datasets, which limits the immediate risk of misuse. We believe that increased transparency about model limitations and failure modes ultimately contributes to safer and more responsible deployment of generative technologies.

## Acknowledgements

Yujin Jeong and Anna Rohrbach gratefully acknowledge support from the hessian.AI Service Center (funded by the Federal Ministry of Research, Technology and Space, BMFTR, grant no. 16IS22091) and the hessian.AI Innovation Lab (funded by the Hessian Ministry for Digital Strategy and Innovation, grant no. S-DIW04/0013/003). This work was supported by the Tübingen AI Center. Arnas Uselis was supported by the International Max Planck Research School for Intelligent Systems (IMPRS-IS). Seong Joon Oh was supported by the Institute for Information & Communications Technology Planning & Evaluation (IITP) grant funded by the Korea government (MSIT) (RS-2019-II190075, Artificial Intelligence Graduate School Program gram(KAIST)). Iro Laina was supported by ERC 101001212-UNION. We also thank the anonymous reviewers for their valuable feedback.

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

# A. Appendix

This supplemental material provides detailed experimental setup (Section A.1), extended experimental results (Section A.2), and qualitative examples (Section A.3). First, we describe experimental details, including those for Figure 1 (b) in main paper, MOSAIC, design choices, training, and evaluation. Then, we present further experimental results, including additional analyses of counting behavior and compositional generalization. Lastly, we show qualitative training examples and generated images.

## A.1. Experimental Setup

### A.1.1. DETAILS FOR FIGURE 1 (B) IN THE MAIN PAPER

**Count frequency.** To understand how data limitations affect diffusion models, we begin by analyzing the frequency of `<count>` + `<object>` phrases in the training dataset used by most diffusion models, LAION-2B (Schuhmann et al., 2022). We first filter captions containing explicit number words. (e.g., "1" or "one") From this pool, we randomly sample 5% for further processing. Because a purely rule-based approach introduces substantial noise, we additionally employ Qwen-8B (Team, 2025) with carefully designed prompts.

> **Prompt:** Find number words (one, two, three, four, five, six, seven, eight, nine, ten) that appear next to or very close to nouns describing countable physical things of any size. ONLY count when: - The number word is adjacent to or within 1-2 words of a concrete noun (like: two dogs, one red car, three small boxes, four tall buildings, two large ships) - The pattern clearly indicates how many X where X is any physical object, structure, vehicle, person, animal, or countable item - Includes small objects (toys, books, cups), medium objects (cars, furniture, appliances), and large objects (buildings, ships, planes, trees) - The context is unambiguous and clearly refers to counting physical things NEVER count when: - Near price/money terms: 'Price: 1 Credit, $5, USD 2 - Part of dates/years: 2019, one year ago - Technical specs: USB 3.0, 4K resolution, iPhone 5 - Ordinals: 1st place, third grade - Measurements: 5 inches, 10 pounds, 3 meters - Abstract concepts: one idea, two reasons - Context is ambiguous or unclear RULE: If its ambiguous, dont count. Only count clear, obvious number+object patterns regardless of object size.

**Counting generation accuracy.** We evaluate SD3-medium (Esser et al., 2024) on counting using prompts derived from CompBench (Huang et al., 2023). Following their object list, we uniformly generate 830 prompts for each target count. Evaluation follows the CompBench protocol, which uses UniDet (Zhou et al., 2022) to assess (i) object presence and (ii) counting accuracy. In our case, we omit (i) and report only (ii), focusing solely on the correctness of the generated object count.

**Spatial relations frequency** We additionally analyze the frequency of spatial relations to show that data scarcity is not limited to counting but also affects relational expressions. To measure the occurrence of spatial relations, we aim to detect patterns of the form `<object>` + `<relation>` + `<object>`. We first filter captions that contain at least one relational term. We group relation phrases into the following categories: **right of**: "right of", "the right", **left of**: "left of", "the left", **above**: "Top of", above", "the Top", **below**: "bottom of", "below", "the bottom", **next to**: "next to", "on side of", "near", **behind**: "behind", "hidden", **in front of**: "in front of".

We use Qwen-8B again with carefully designed prompts (shown below) to validate and refine the extracted relations.

> **Prompt:** Count spatial position phrases in this caption. Look for these EXACT phrases that describe object locations: ONLY count if you see these EXACT words describing WHERE objects are positioned: • 'right of', 'to the right', 'on the right' - objects positioned to the right • 'left of', 'to the left', 'on the left' - objects positioned to the left • 'above', 'on Top of', 'over' - objects positioned higher • 'below', 'under', 'beneath' - objects positioned lower • 'behind', 'in back of' - objects positioned in back • 'in front of', 'in the front' - objects positioned in front • 'next to', 'beside', 'near' - objects positioned adjacent CRITICAL: Only count if these phrases are actually present in the text describing object positions. DO NOT count words that are not explicitly in the caption. DO NOT make assumptions about implied positions. Instructions: 1. Read the caption carefully 2. Look for the EXACT spatial phrases listed above 3. Count ONLY what is explicitly written 4. If none found, all counts should be 0

Figure A.1 shows that some spatial relation terms appear far more frequently than others, indicating a substantial imbalance in the Laion2B caption. This confirms that our concept imbalance design is not only relevant for analyzing counting, but also extends more broadly to spatial relations.

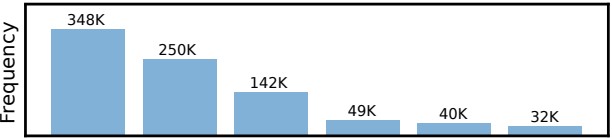

*Figure A.1.* **Frequency of spatial relation concepts in the LAION-2B captions.** The distribution is highly imbalanced, with "in front of" appearing more than 10 times as often as "left of" or "right of", highlighting strong biases in relational supervision.

### A.1.2. MOSAIC DESIGN DETAILS

**Details of MOSAIC.** MOSAIC builds on the 3D scene assets from COMFORT-BALL (Zhang et al., 2024), originally developed for evaluating spatial reasoning in vision–language models, while maintaining photorealistic rendering quality at 512×512 resolution. We restructure and extend these assets to enable systematic control over compositional multi-object concepts and dataset configurations. Additionally, we incorporate cube assets from MULTIMODAL3DIDENT (Daunhawer et al., 2023).

Our goal is to construct a minimally complex environment that isolates multi-object reasoning without confounding visual challenges. To this end, we fix the camera to a top-down view, introduce no occlusions between objects, and deliberately limit objects' diversity. This ensures that performance differences reflect conceptual understanding rather than low-level visual variation.

Importantly, MOSAIC is designed as a diagnostic framework rather than a realistic generative benchmark. The same assets can be easily extended to include occlusions, distractors, varied viewpoints, or scene complexity, enabling future work to study harder settings while maintaining compatibility with our controlled base environment.

**Grid-based configurations.** To reduce spatial complexity, we introduce a grid-based variant of MOSAIC. The grid partitions the scene into ten angular sectors of 18° each, separated by 18° gaps, matching the discretization used for spatial relations. Formally, the sectors cover the following angular intervals (measured counter-clockwise from the reference axis): $(0°, 18°), (36°, 54°), (72°, 90°),$ $(108°, 126°), (144°, 162°), (180°, 198°), (216°, 234°),$ $(252°, 270°), (288°, 306°), (324°, 342°)$

For the Counting task, objects are placed within fixed sectors based on the target count: a single object appears in sector 1; two objects appear both in sectors 1 and 2; and so on. Objects may vary within each assigned sector but cannot move outside it. For the Spatial Relations task, the BROWN sphere serves as a fixed reference at the center of the grid, and the second object is placed in a sector corresponding to the target angular relation. Qualitative examples are provided in Figure A.28.

**Class imbalance details.** For the *skewed* setting, we construct datasets with controlled degrees of class imbalance while keeping the skewness pattern consistent across different dataset sizes. For example, in a 100k dataset, the *skewed* distribution allocates (22,550, 17,950, 14,350, 11,450, 9,150, 7,300, 5,850, 4,650, 3,750, 3,000) samples across the ten classes. For 50k, 10k, and 2k datasets, we use proportional distributions: 50k: (11,275, 8,975, 7,175, 5,725, 4,575, 3,650, 2,925, 2,325, 1,875, 1,500),

10k: (2,255, 1,795, 1,435, 1,145, 915, 730, 585, 465, 375, 300), and 2k: (451, 359, 287, 229, 183, 146, 117, 93, 75, 60).

**Compositional generalization design.** For compositional generalization, we progressively remove diagonals from the concept-pair matrix, following the previous work's design choice (Uselis et al., 2025). If one diagonal is designated as unseen, only the first diagonal is removed; if three diagonals are unseen, the first three diagonals are removed, and so on. This procedure allows us to systematically control how many concept pairs are never observed during training while ensuring that *all individual concepts remain fully observed*. Tables A.1, A.2, A.3, and A.4 define the diagonal indices for the Attribution, Spatial Relations, Counting and Objects tasks, respectively.

In the main paper, we have removed $k \in \{0, 1, 3, 5, 8\}$ diagonals and resample the remaining compositions to keep dataset sizes comparable. For example, in the 100k setting, removing one diagonal yields 99,990 samples, and removing three diagonals yields 99,960 samples. For five and eight diagonals, we match the full 100k samples.

*Table A.1.* **Compositional configuration for Sphere Color × Cube Color (Attribution).** Numbers indicate diagonal indices used to determine which concept pairs are removed in unseen-composition settings. Lower numbers correspond to diagonals removed first. Rows represent sphere colors and columns represent cube colors.

| | RED | GREEN | BLUE | YELLOW | PURPLE | ORANGE | CYAN | GRAY | WHITE | BLACK |
|---|---|---|---|---|---|---|---|---|---|---|
| RED | 1 | 2 | 3 | 4 | 5 | 6 | 7 | 8 | 9 | 10 |
| GREEN | 10 | 1 | 2 | 3 | 4 | 5 | 6 | 7 | 8 | 9 |
| BLUE | 9 | 10 | 1 | 2 | 3 | 4 | 5 | 6 | 7 | 8 |
| YELLOW | 8 | 9 | 10 | 1 | 2 | 3 | 4 | 5 | 6 | 7 |
| PURPLE | 7 | 8 | 9 | 10 | 1 | 2 | 3 | 4 | 5 | 6 |
| ORANGE | 6 | 7 | 8 | 9 | 10 | 1 | 2 | 3 | 4 | 5 |
| CYAN | 5 | 6 | 7 | 8 | 9 | 10 | 1 | 2 | 3 | 4 |
| GRAY | 4 | 5 | 6 | 7 | 8 | 9 | 10 | 1 | 2 | 3 |
| WHITE | 3 | 4 | 5 | 6 | 7 | 8 | 9 | 10 | 1 | 2 |
| BLACK | 2 | 3 | 4 | 5 | 6 | 7 | 8 | 9 | 10 | 1 |

*Table A.2.* **Compositional configuration for Angle × Color (Spatial Relations).** Entries indicate diagonal indices used to determine which combinations are held out. Removing earlier diagonals increases compositional difficulty while keeping all individual concepts observed.

| | RED | GREEN | BLUE | YELLOW | PURPLE | ORANGE | CYAN | GRAY | WHITE | BLACK |
|---|---|---|---|---|---|---|---|---|---|---|
| angle1 $(0°, 18°)$ | 1 | 2 | 3 | 4 | 5 | 6 | 7 | 8 | 9 | 10 |
| angle2 $(36°, 54°)$ | 10 | 1 | 2 | 3 | 4 | 5 | 6 | 7 | 8 | 9 |
| angle3 $(72°, 90°)$ | 9 | 10 | 1 | 2 | 3 | 4 | 5 | 6 | 7 | 8 |
| angle4 $(108°, 126°)$ | 8 | 9 | 10 | 1 | 2 | 3 | 4 | 5 | 6 | 7 |
| angle5 $(144°, 162°)$ | 7 | 8 | 9 | 10 | 1 | 2 | 3 | 4 | 5 | 6 |
| angle6 $(180°, 198°)$ | 6 | 7 | 8 | 9 | 10 | 1 | 2 | 3 | 4 | 5 |
| angle7 $(216°, 234°)$ | 5 | 6 | 7 | 8 | 9 | 10 | 1 | 2 | 3 | 4 |
| angle8 $(252°, 270°)$ | 4 | 5 | 6 | 7 | 8 | 9 | 10 | 1 | 2 | 3 |
| angle9 $(288°, 306°)$ | 3 | 4 | 5 | 6 | 7 | 8 | 9 | 10 | 1 | 2 |
| angle10 $(324°, 342°)$ | 2 | 3 | 4 | 5 | 6 | 7 | 8 | 9 | 10 | 1 |

### A.1.3. TRAINING DETAILS.

**Training diffusion models.** All diffusion models are trained on four A100 GPUs with a batch size of 512 per

*Table A.3.* **Compositional configuration for Count × Color (Counting).** Diagonal indices specify the order in which concept pairs are removed to create unseen composition settings. Higher unseen-diagonal counts correspond to harder compositional generalization.

|         | RED | GREEN | BLUE | YELLOW | PURPLE | ORANGE | CYAN | GRAY | WHITE | BLACK |
|---------|-----|-------|------|--------|--------|--------|------|------|-------|-------|
| count1  | 1   | 2     | 3    | 4      | 5      | 6      | 7    | 8    | 9     | 10    |
| count2  | 10  | 1     | 2    | 3      | 4      | 5      | 6    | 7    | 8     | 9     |
| count3  | 9   | 10    | 1    | 2      | 3      | 4      | 5    | 6    | 7     | 8     |
| count4  | 8   | 9     | 10   | 1      | 2      | 3      | 4    | 5    | 6     | 7     |
| count5  | 7   | 8     | 9    | 10     | 1      | 2      | 3    | 4    | 5     | 6     |
| count6  | 6   | 7     | 8    | 9      | 10     | 1      | 2    | 3    | 4     | 5     |
| count7  | 5   | 6     | 7    | 8      | 9      | 10     | 1    | 2    | 3     | 4     |
| count8  | 4   | 5     | 6    | 7      | 8      | 9      | 10   | 1    | 2     | 3     |
| count9  | 3   | 4     | 5    | 6      | 7      | 8      | 9    | 10   | 1     | 2     |
| count10 | 2   | 3     | 4    | 5      | 6      | 7      | 8    | 9    | 10    | 1     |

*Table A.4.* **Compositional configuration for Object × Object (object co-occurence).** Diagonal indices specify the order in which concept pairs are removed to create unseen composition settings.

|            | Bicycle | Couch | Chair | Dog | Bed | Laptop | Bench | Sophia | Basketball | Horse |
|------------|---------|-------|-------|-----|-----|--------|-------|--------|------------|-------|
| Bicycle    | 1       | 2     | 3     | 4   | 5   | 6      | 7     | 8      | 9          | 10    |
| Couch      | 10      | 1     | 2     | 3   | 4   | 5      | 6     | 7      | 8          | 9     |
| Chair      | 9       | 10    | 1     | 2   | 3   | 4      | 5     | 6      | 7          | 8     |
| Dog        | 8       | 9     | 10    | 1   | 2   | 3      | 4     | 5      | 6          | 7     |
| Bed        | 7       | 8     | 9     | 10  | 1   | 2      | 3     | 4      | 5          | 6     |
| Laptop     | 6       | 7     | 8     | 9   | 10  | 1      | 2     | 3      | 4          | 5     |
| Bench      | 5       | 6     | 7     | 8   | 9   | 10     | 1     | 2      | 3          | 4     |
| Sophia     | 4       | 5     | 6     | 7   | 8   | 9      | 10    | 1      | 2          | 3     |
| Basketball | 3       | 4     | 5     | 6   | 7   | 8      | 9     | 10     | 1          | 2     |
| Horse      | 2       | 3     | 4     | 5   | 6   | 7      | 8     | 9      | 10         | 1     |

GPU, using AdamW (Loshchilov & Hutter, 2017) for the optimizer. Training takes approximately five hours including validation. During validation, we generate 50 samples per prompt and compute validation accuracy. Generation quality is monitored every 500 steps, and unless otherwise stated, we report results from the best validation checkpoint. Based on these results, we adopt 0.0001 as the default learning rate for all experiments. Unless otherwise noted, models are trained for 20k steps. The only exception is the Counting experiment in Section 6 of the main paper, where additional training is required for the accuracy to reach saturation.

All images are resized to 128×128 resolution. We select the learning rate through a sweep over 0.001, 0.0001, 0.00001 using 20k training steps on the Counting (100k, Uniform) dataset. This task exhibits the highest sensitivity to hyperparameters, making it a suitable benchmark for LR selection. Table A.5 shows the corresponding best validation accuracies for three model sizes (40M, 90M, 200M). For our baseline 90M model, 0.0001 yields the most reliable performance (0.962), whereas 0.001 and 0.00001 show significant degradation. In particular, with 0.001 the model briefly reaches 0.486 accuracy at step 1,000 before the loss becomes unstable, indicating that this learning rate is too large for consistent training. We use a constant learning-rate schedule in all experiments. We have also tested other schedules (e.g., cosine decay, linear), but observe noticeable differences are observed in performance.

*Table A.5.* **Best validation accuracy on the Counting (100k, Uniform) dataset**. The 0.0001 learning rate yields the most stable performance for the 90M baseline model.

| Learning rate / Model size | 40M   | **90M (Ours)** | 200M  |
|----------------------------|-------|----------------|-------|
| 0.001                      | 0.988 | 0.486          | 0.972 |
| **0.0001**                 | 0.956 | **0.962**      | 0.978 |
| 0.00001                    | 0.86  | 0.868          | 0.886 |

**Unet-based diffusion backbone training objectives.** Given an image and one-hot vector condition pair $(x, y)$, the VAE encodes $x$ into a latent $z$. At diffusion timestep $t$, noise $\epsilon$ is added to obtain $z_t$, and the U-Net predicts the injected noise conditioned on the encoded representation $c(y)$:

$$\mathcal{L}(z, y) = \mathbb{E}_{t,\epsilon}\left[\|\epsilon - \epsilon_\Theta(z_t, t, c(y))\|^2\right], \qquad \text{(A.1)}$$

where $c(\cdot)$ is the condition encoder and $\epsilon_\Theta(\cdot)$ denotes the U-Net noise prediction network. Both are jointly trained.

**DiT backbone Architectures.** Recent text-to-image (T2I) generation models (Esser et al., 2024; Yang et al., 2024a) often adopt Diffusion Transformers (DiT) architectures (Peebles & Xie, 2023) trained with rectified flow objectives (Lipman et al., 2023).

We adopt the DiT architecture from SD3 (Esser et al., 2024) and reduce the model size to approximately 90M parameters to closely match the capacity of our UNet-based baseline. To ensure comparable image quality across architectures, we fix the VAE to the one used in SD2, which is also used for all UNet-based experiments in this work.

In the original SD3 design, text embeddings are injected through two pathways: (i) token-level embeddings are provided to the cross-attention layers for conditional generation, and (ii) pooled text embeddings are added to the timestep embeddings and combined with the model's global conditioning. In our setting, since we do not use pooled text embeddings for the structured conditional inputs, we only provide conditional embeddings to the attention layers and do not add any additional conditioning to the timestep embeddings.

Finally, since the SD3 architecture requires sufficiently large embedding dimensions for normalization layers, we increase the depth of the conditional encoder to produce higher-dimensional embeddings compatible with the DiT blocks.

**DiT-based diffusion backbone training objectives.** Similar to Equation A.1, given an image and one-hot vector condition pair $(x, y)$, the VAE encodes $x$ into a latent $z$. At diffusion timestep $t$, noise $\epsilon$ is added to obtain $z_t$, and the

DiT predicts the injected noise conditioned on the encoded representation $c(y)$:

$$\mathcal{L}_{\text{RF}}(z, y) = \mathbb{E}_{t,\epsilon}\left[w_t \left\| \epsilon_\Theta(z_t, t, c(y)) - \epsilon \right\|^2\right], \quad \text{(A.2)}$$

where $c(\cdot)$ is the condition encoder, $\epsilon_\Theta(\cdot)$ denotes the DiT noise prediction network, and $w_t$ is the time-dependent weighting term defined by the rectified flow (CFM) formulation following Esser et al. (2024). Both condition encoder and DiT are jointly trained.

**Training classifiers.** Classifier models are trained on a single L40S GPU using the AdamW optimizer with cross-entropy loss. These classifiers are used to evaluate the diffusion models' generated samples. Input images are resized to $128 \times 128$ to match the resolution of the generated samples. To improve robustness, we apply data augmentation with 0.3 probability during training, depending on the task. VAE reconstruction is applied as a general augmentation across all settings. For the counting/relation classifier, we apply color jitter and intensity augmentation but avoid spatial transformations such as cropping, which would alter the object count. For the color classifier, we apply stronger blur-based augmentations to increase invariance to local texture noise. Training is early-stopped if the validation loss does not improve for 25 epochs.

### A.1.4. EVALUATION DETAILS

**Classifiers.** Our pretrained classifiers predict: (i) 20 count classes for Counting (we extend beyond 10 objects since diffusion models often generate more than ten spheres), (ii) 10 spatial-relation classes for Spatial relations, and (iii) 100 color-pair classes for Attribution.

For concept generalization (RQ1), we report per-class accuracy for each task and a memorization rate. For compositional generalization (RQ2), we report joint accuracy for Counting and Spatial Relations, obtained by additionally training a 10-class color classifier to evaluate the combined (count/spatial relation, color) output.

To ensure reliable evaluation, all classifiers are trained on an extended version of MULTI-COMFORT containing between 100k and 1M images, using 5% for validation and 5% for testing. As a lower-bound sanity check, we evaluate VAE reconstructions (without diffusion) on 5k-20k random samples. These reconstructions achieve near-perfect accuracy, confirming that the classifiers are reliable for evaluation (See Table A.6).

*Table A.6.* **Classification accuracy on VAE reconstructions**. The near-perfect scores indicate that classifier predictions are reliable.

|  | Accuracy | Accuracy (Color) |
|---|---|---|
| Counting | 0.9997 | 0.9996 |
| Attribution | 0.9998 | - |
| Spatial relations | 1.0 | 0.9996 |

**Evaluation metrics.** We report two main metrics: *Accuracy*, which serves as our primary evaluation measure, and *Memorization rate*, which quantifies proximity to training examples.

*Accuracy* is defined as

$$\text{Acc} = \frac{1}{N} \sum_{i=1}^{N} \mathbb{1}\left[f_{\text{clf}}(x_i) = y_i\right], \quad \text{(A.3)}$$

where $x_i$ is the generated image, $y_i$ is the ground-truth conditioning label (count, relation, or color-pair for attribution), $f_{\text{clf}}(\cdot)$ denotes the pretrained classifier, $\mathbb{1}[\cdot]$ is the indicator function.

*Memorization rate* follows the definition introduced in prior work (Bonnaire et al., 2025). A generated sample $\mathbf{x}_\tau$ is considered memorized if

$$\mathbb{E}_{\mathbf{x}_\tau}\left[\frac{\|\mathbf{x}_\tau - \mathbf{a}^{\mu_1}\|_2}{\|\mathbf{x}_\tau - \mathbf{a}^{\mu_2}\|_2}\right] < k, \quad \text{(A.4)}$$

where $\mathbf{a}^{\mu_1}$ and $\mathbf{a}^{\mu_2}$ are the nearest and second-nearest neighbors of $\mathbf{x}_\tau$ in the training set in the $L_2$ pixel distance sense and we have set $k = 1/3$ following the previous work.

**Evaluation protocol.** Across all settings, we generate 50 samples per condition using DDIM sampling without classifier-free guidance.

### A.2. Additional Analysis

For the counting analysis, we focus on the UNet backbone unless otherwise stated, since the accuracy degradation is more severe than in DiT architectures.

### A.2.1. COUNTING BEHAVIOR ANALYSIS

**Effect of model size.** As observed in Section 5 (main paper), counting accuracy drops sharply for small dataset sizes, regardless of the degree of skewness, and gradually recovers as the dataset size increases. Only when the dataset is sufficiently large does the model reliably generate the correct number of objects, independent of the training distribution. Importantly, this behavior cannot be attributed to model capacity: both smaller and larger models exhibit the same trend, as shown in Figure A.2.

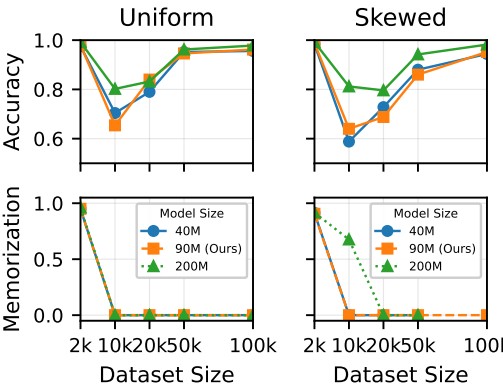

*Figure A.2.* **Influence of model size on Counting accuracy.** Larger or smaller models do not mitigate the failure at moderate dataset sizes; all model sizes follow the same trend.

**Memorization rate.**  Figure A.3 reports the memorization rate under the default concept generalization setting, while Figure A.4 shows the corresponding results under increased scene complexity. At the smallest dataset sizes (e.g., 2k), all three tasks exhibit near-100% memorization, indicating strong overfitting. As the dataset scale increases, memorization gradually decreases across all tasks. Notably, all tasks exhibit an intermediate regime in which memorization is no longer feasible.

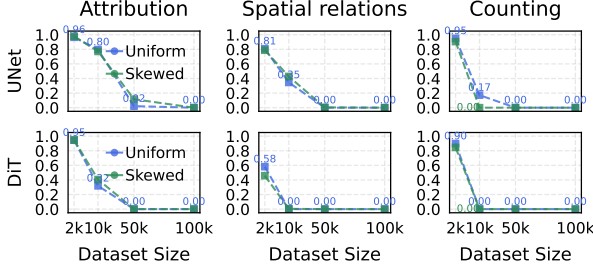

*Figure A.3.* **Memorization rate under the default setting.** The x-axis denotes dataset size and the y-axis denotes memorization rate. Memorization is near 100% at small dataset sizes but gradually decreases as the dataset scale increases.

**Training loss.**  As shown in Figure A.5, training loss consistently decreases across all dataset sizes and model architectures, indicating stable optimization behavior.

**High object counts collapse first.**  We first examine per-class accuracy at the best validation checkpoint for two dataset sizes (10k and 100k). In Fig. A.6 left, we observe that higher counts are more difficult: with 10k samples, accuracy is sharply skewed toward lower counts, while counts 6–10 are substantially worse: single object scenes are generated with 100%, while scenes with ten objects are generated

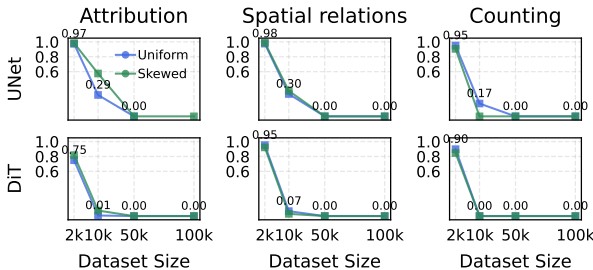

*Figure A.4.* **Memorization rate under increased scene complexity.** The x-axis denotes dataset size and the y-axis denotes memorization rate. Compared to the default setting, memorization decreases more rapidly as dataset size increases.

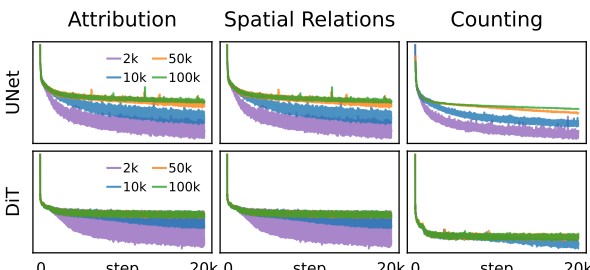

*Figure A.5.* **Training loss across dataset sizes and architectures.** Training loss decreases consistently during training for all settings, suggesting stable optimization.

only with 44% accuracy. This indicates that generating many distinct object instances is harder than generating a few, even in a uniform setting.

The right-side plots show how per-class accuracy evolves during training. At 10k dataset, we observe a progressive collapse toward under-counting as training continues. Only when the dataset is sufficiently large (100k) does accuracy stabilize across all count values; yet even there counts 9 and 10 are slightly harder to generate the others. This means that the complexity matters even in simple datasets, which means that multiple-objects leads to more failures in the generation.

**Pixel-space distance to training samples.**  To further investigate selective overfitting, we analyze which labels contribute most to memorization. For each successfully generated image, we compute its pixel-space distance to the nearest training image and plot the distribution of these minimum distances as a histogram. Smaller distances indicate that a generated image closely resembles a specific training sample.

Figure A.7 shows memorization behavior at the final training step for dataset sizes 10k (Top) and 50k (Bottom), with the red line indicating the mean distance across all labels. In both cases, lower-count classes exhibit substantially smaller pixel distances. Even at the 50k scale, below-average dis-

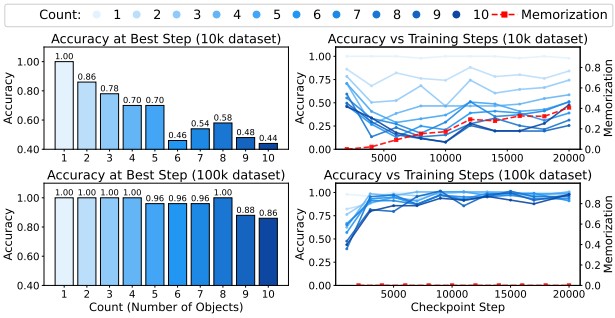

*Figure A.6.* **Per-class accuracy for counting.** (top) 10k dataset: Higher object counts suffer the most. Accuracy peaks early during training but later degrades, with partial recovery driven by memorization rather than genuine generalization. (bottom) 100k dataset: Higher counts are still more difficult but remain much more stable. All count classes converge to high accuracy, and memorization stays near zero throughout training.

tances are concentrated almost exclusively among counts $< 5$, suggesting that low-count samples begin to be memorized first, while higher counts remain harder to memorize.

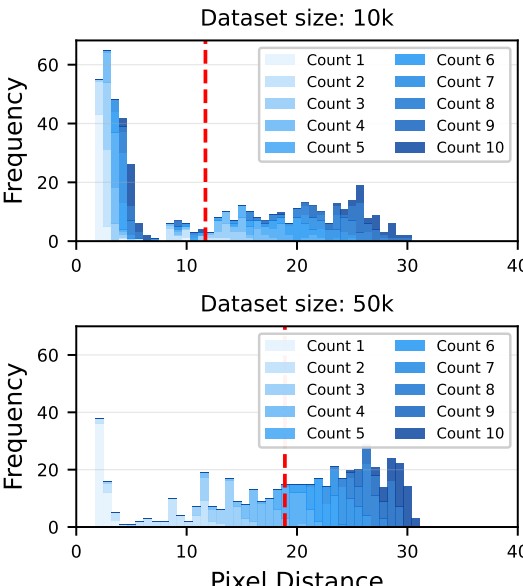

*Figure A.7.* **Pixel-space distance to nearest training samples by count label.** Pixel-distance histograms to nearest training samples for 10k (Top) and 50k (Bottom) datasets. Lower count labels exhibit smaller distances, indicating initial memorization, while higher counts remain far from training samples.

**Confusion matrix at 10k dataset size.** As discussed in Section 5 of the main paper, the 10k dataset exhibits a progressive collapse toward *under-counting* during training: although the model initially produces the correct number of objects, it gradually shifts toward predicting fewer objects than the ground truth.

Figure A.8 (top) compares confusion matrices at the best validation step (2,000) and the final training step (20,000). At early steps, predictions are largely diagonal, indicating correct counts across most classes. However, by the end of training, predictions skew heavily toward lower count classes, demonstrating a systematic bias toward generating fewer instances over time.

Importantly, this collapse is not accompanied by visible degradation in image quality. As shown in Figure A.8 (bottom), generated samples remain visually sharp and diverse. Thus, the decline in accuracy does not reflect training instability or loss of texture fidelity, but rather the loss of count information specifically, while other aspects of the generation remain intact.

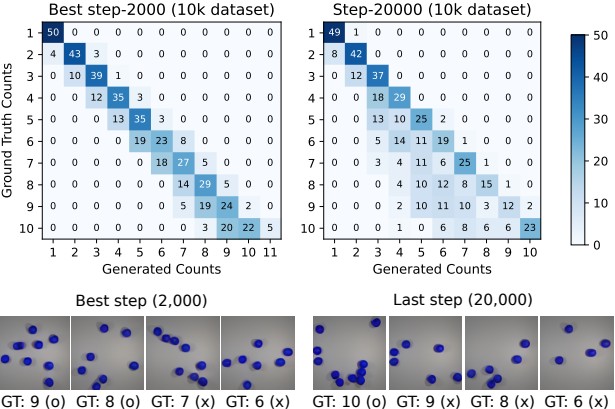

*Figure A.8.* **Confusion matrices for the Counting task at dataset size 10k.** (Top) Confusion matrices at 2k in the left and 20k in the right. Training induces a systematic shift toward lower predicted counts. (Bottom) Despite this collapse, visual quality remains intact, indicating loss of count information rather than general generation failure.

**Validation loss curves.** Figure A.9 (left) plots validation loss across dataset sizes. Validation loss increases for all settings except 100k, indicating that models trained on 2k, 10k, and 50k overfit early, whereas larger datasets mitigate overfitting. Despite this overfitting, validation accuracy on Counting decreases (right) on 10k and 50k, reflecting a form of *selective overfitting* in which the model continues to optimizing non-count-related information.

**Condition embeddings.** We investigate whether condition embeddings collapse when the dataset is small. Figure A.10 (top) visualizes the PCA projection of the count embeddings (two principal components) at the final training step (20,000) for dataset sizes 10k, 50k, and 100k (left to right). We observe that embeddings are substantially more collapsed in the 10k setting, whereas they remain more separable at 50k and 100k, suggesting that the model loses discriminatory capacity over count labels at 10k.

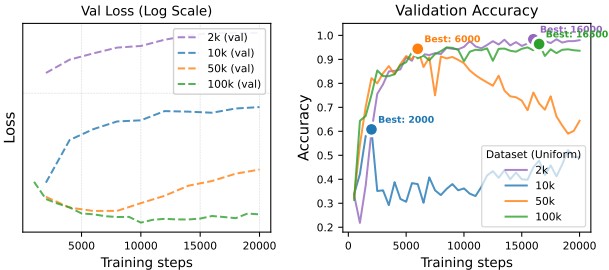

*Figure A.9.* **Validation loss and accuracy for Counting across dataset sizes.** (Left) Validation loss increases for all dataset sizes except 100k, indicating overfitting in smaller datasets (2k, 10k, 50k). (Right) Unlike 2k and 50k, validation accuracy on 10k and 50k dataset size peak early then deteriorate.

To mitigate this issue, we add auxiliary losses only to the condition encoder during training on the 10k dataset: (i) a cross-entropy classification loss (green), and (ii) a contrastive InfoNCE loss (Oord et al., 2018) (orange). We also evaluate a frozen condition encoder (red) that is pretrained with cross-entropy loss and not jointly optimized with the diffusion model. However, as shown in Figure A.10 (bottom), the embedding collapse persists even with these objectives, indicating that the issue does not stem solely from inadequate supervision of the condition encoder.

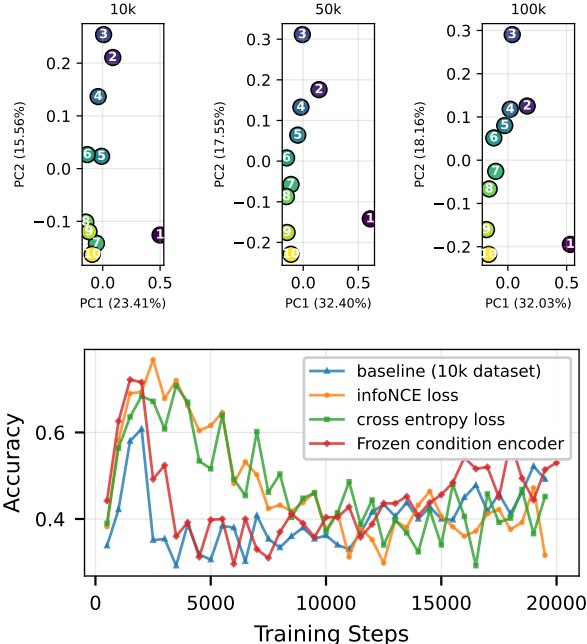

*Figure A.10.* **Condition embedding collapse under small data.** (Top) PCA visualization of count-conditioned embeddings at the final training step for datasets of size 10k, 50k, and 100k. The 10k setting shows collapse across count classes, whereas 50k and 100k maintain clear separation. (Bottom) Validation accuracy across training steps for the 10k dataset shows that collapse persists even when additional classification losses are applied or when using a frozen encoder that was already trained with classification.

**Heterogeneous vs. identical objects.** We construct an additional setting in which object attributes (shape, size, and color) are randomly varied, resulting in heterogeneous object compositions (examples shown in Fig.A.11). We evaluate counting performance under this setting and compare it with the original setup using identical objects. The results are summarized in Fig.A.12. These results suggest that counting relies on consistent object representations, which become harder to maintain as object diversity increases, leading to more severe failures. We further observe that the collapse persists in the heterogeneous setting, as shown in Fig. A.13.

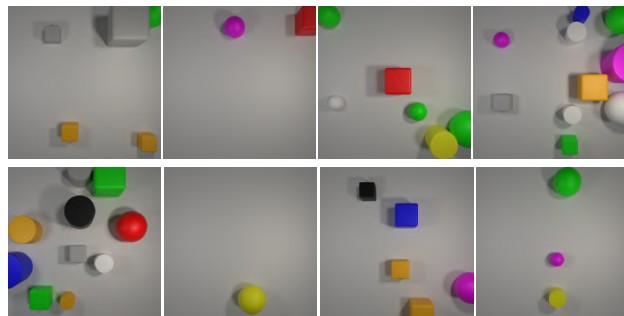

*Figure A.11.* **Examples of heterogeneous objects for Counting.** Unlike the original Counting setup with identical spheres, this setting randomly varies object attributes such as shape, size, and color while keeping the target count fixed.

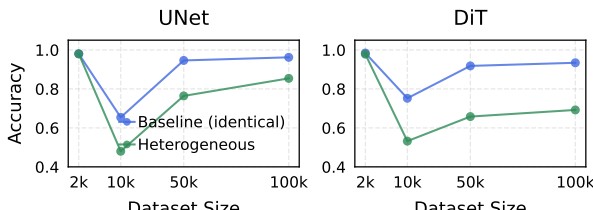

*Figure A.12.* **Counting performance of identical vs. heterogeneous objects.** We compare the original Counting setup with identical objects against the heterogeneous setting, where object attributes are randomly varied. Heterogeneous objects lead to lower counting accuracy, suggesting that object diversity makes numerosity estimation more difficult.

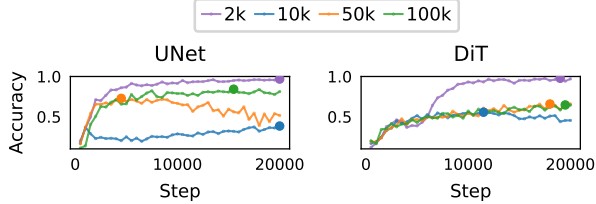

*Figure A.13.* **Counting accuracy trajectories over training steps with heterogeneous objects.** The trajectories show that the performance collapse observed in the original Counting setup also persists when object attributes are randomly varied.

### A.2.2. Compositional Generalization Analysis

**Effect of grid layouts on compositional generalization.**
Figure A.18 shows accuracy for Spatial relations and Counting on unseen compositions. The left column reports joint accuracy (our primary metric), the middle column reports relation/count accuracy, and the right column reports color accuracy. We observe moderate improvements in Spatial realtions and Counting under the grid setting; however, once roughly half of the diagonals are held out, color accuracy declines due to a trade-off introduced by the reduced spatial complexity. Moreover, performance continues to deteriorate as more compositions are withheld, indicating that spatial priors alone are insufficient to enable compositional generalization.

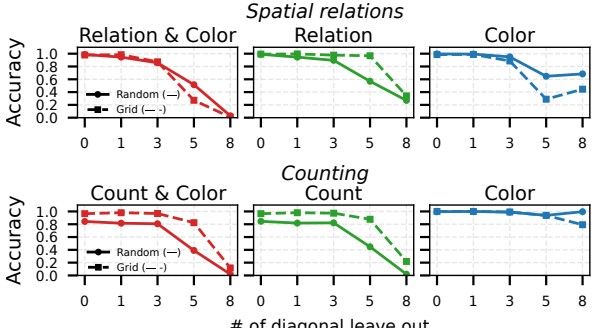

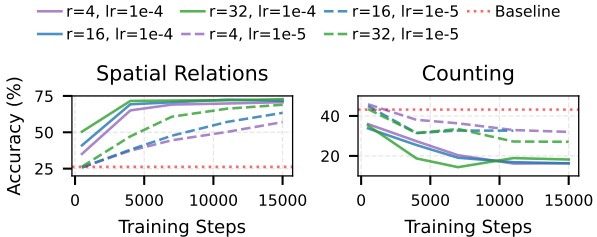

*Figure A.14.* **Effect of lowering spatial complexity on compositional settings.** Columns report Joint accuracy, task-specific accuracy, and Color accuracy on unseen compositions (diagonals). Reducing scene complexity leads to modest improvements in Spatial relation and Counting accuracy, but simultaneously degrades Color accuracy, resulting in overall performance comparable to the non-grid setting.

**LoRA ablation study for fine-tuning SD3.** Across different LoRA ranks $r$ and learning rates, we observe consistent trends when fine-tuning SD 3 (Esser et al., 2024): spatial relation accuracy improves, whereas counting accuracy deteriorates.

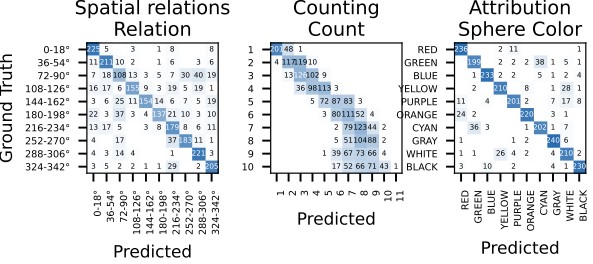

*Figure A.15.* **LoRA ablation across ranks and learning rates.** Results are consistent across hyperparameter settings: spatial relation accuracy improves with fine-tuning, while counting accuracy degrades.

**Compositional generalization with Unet backbone** In addition to the analysis on compositional generalization with DiT, where models fail to generalize to unseen compositions

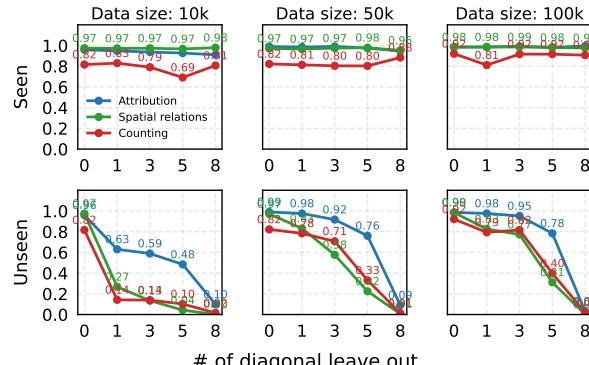

*Figure A.16.* **Compositional generalization on dataset size and the number of unseen compositions on Unet.** (Top) For seen compositions, Attribution and Spatial relations remain stable across all dataset sizes, while Counting improves noticeably as the dataset size increases. (Bottom) For unseen compositions, performance drops rapidly as the dataset size decreases or the number of held-out compositions increases.

*Figure A.17.* **Confusion matrix on unseen diagonals when half of the compositions (5 diagonals) are unseen.** Compared to Attribution and Counting, Spatial relations show no clear error pattern. This indicates that spatial relations are highly fragile, and easily break when compositions are unseen.

when only a small subset of compositions is observed during training.

Following Section 6 of the main paper, we evaluate compositional generalization for UNet architectures. Figure A.16 reports accuracy on unseen compositions as the number of diagonals held out during training increases. Performance drops sharply once more than half of the compositions are unseen, indicating severe failures in compositional generalization. Attribution remains comparatively robust, whereas Counting and Spatial Relations collapse under large composition gaps. This behavior is further supported by the confusion matrices in Figure A.17, which exhibit widespread misclassification patterns. Introducing a spatial grid layout (Figure A.18) provides only limited improvement and does not recover compositional generalization. Overall, the failure mode remains unchanged: increasing dataset scale alone is insufficient, and compositional generalization does not reliably emerge even with a stronger architecture and improved training objectives.

**Is a compositionally broken text encoder responsible for failures in compositional generation?** Since our model

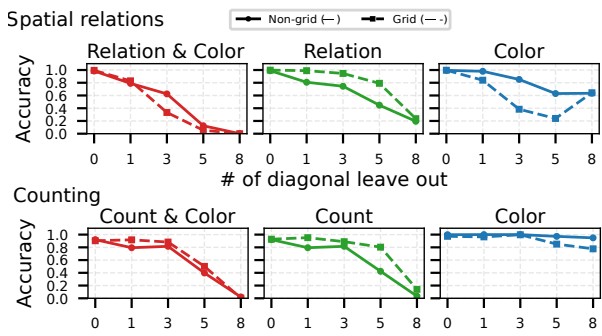

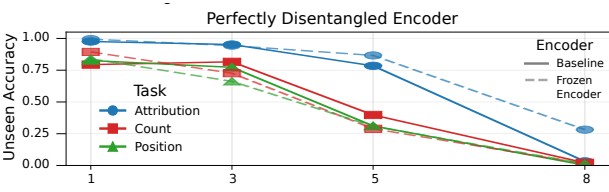

*Figure A.18.* **Effect of lowering spatial complexity on compositional settings.** Columns report Joint accuracy, task-specific accuracy, and Color accuracy on unseen compositions (diagonals). Reducing scene complexity leads to modest improvements in Spatial relation and Counting accuracy, but simultaneously degrades Color accuracy, resulting in overall performance comparable to the non-grid setting.

*Figure A.19.* **Effect of condition encoder disentanglement on compositional generalization.** Dashed lines correspond to results obtained using a frozen, disentangled condition encoder, while solid lines correspond to the baseline, where the encoder is jointly trained with the diffusion model. The performance gap remains small, indicating limited benefit from disentangling the condition embeddings alone.

adopts a cross-attention-based conditioning mechanism similar to real-world text-to-image diffusion models, we examine whether improving the condition encoder alone is sufficient to recover compositional generalization. Several prior works (Huang et al., 2024; Tong et al., 2023; Zarei et al.; Toker et al., 2024) argue that the text encoder plays a critical role in compositional generation. Here, we define a *compositionally broken* encoder as one in which token representations are not disentangled across concepts (e.g., "red" and "apple" are not cleanly separable in the embedding space). To test this hypothesis, we replace the jointly trained condition encoder with a frozen, pretrained encoder trained using cross-entropy supervision, which produces more disentangled condition embeddings. Figure A.19 shows that this disentangled encoder provides only marginal improvements in compositional accuracy and does not substantially recover compositional generalization. This suggests that failures in compositional generation cannot be attributed solely to deficiencies in the condition encoder, but instead reflect limitations in the diffusion model's ability to bind and recombine concepts.

**Compositional accuracy saturates.** As shown in Figure A.20, validation accuracy plateaus for both seen and unseen compositions, indicating that extended training does not lead to further improvements in compositional generalization.

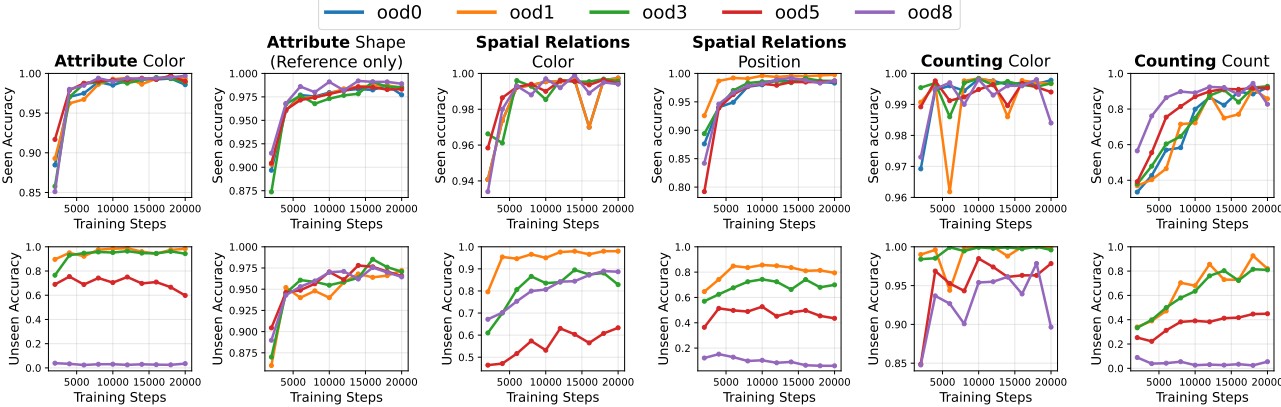

*Figure A.20.* **Validation accuracy dynamics across the number of unseen diagonals.** (Top) Accuracy on seen compositions. (Bottom) Accuracy on unseen compositions. Both curves plateau, indicating that longer training does not improve compositional generalization.

A.2.3. ANALYSIS ON MORE REALISTIC SETUP

**Effect of text-based conditioning.** Our original design choice of using one-hot condition encoding was intended to isolate the effect of compositional generalization without introducing additional compositional capacity from the encoder itself. To this end, we jointly train the condition encoder with the diffusion model. However, such a simplified encoding may not fully capture real-world settings.

To address this, we conduct additional experiments using a CLIP-B/16 text encoder (Radford et al., 2021) as the conditioning module for a DiT backbone, with the encoder kept frozen during training. We design prompts aligned with MOSAIC factors: Attribution uses prompts such as "A photo of a black sphere and a red cube"; Counting uses prompts such as "A photo of ten blue spheres"; and Spatial Relations uses prompts such as "A photo of the red sphere at an angle of 222° from a brown sphere, measured from the rightward direction". In practice, we use angle ranges such as 215°–234°.

The results in Figure A.21 show a slight improvement in concept generalization at the 10k dataset size when using CLIP text embeddings as conditioning, compared to the one-hot encoding baseline. We also evaluate compositional generalization in Figure A.22, where the gains are more noticeable. To further isolate the effect of the encoder in the compositional generalization setting, we include an additional baseline with a frozen condition encoder trained separately using one-hot inputs with BCE loss, denoted as Baseline (frozen)" in the figure. This corresponds to the experiments in Appendix Figure A.19 using a UNet backbone. We find that its performance is comparable to CLIP-based conditioning. Importantly, the overall trends remain unchanged: (i) counting remains difficult in low-data regimes, and (ii) performance consistently degrades as more compositions are held out. These results suggest that the observed limitations are not primarily due to the simplicity of the

condition encoding.

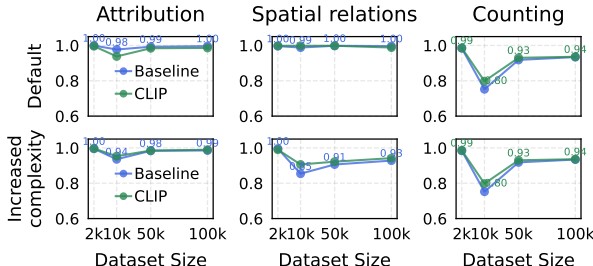

*Figure A.21.* **Concept generalization with a DiT architecture under one-hot conditioning and text embeddings.**

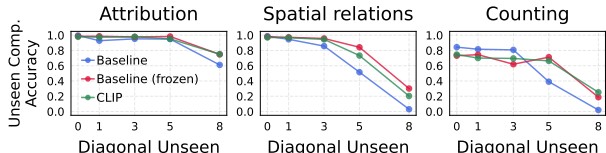

*Figure A.22.* **Compositional generalization with a DiT architecture under different conditioning strategies.** We compare a jointly trained condition encoder with one-hot inputs, a separately trained frozen condition encoder with one-hot inputs, and CLIP-based text embeddings.

**3D Spatial Relations.** We introduce a more realistic geometric setting by varying camera perspectives, which induces depth, scale changes, and occlusions (Figure A.23). While constructing explicit 3D relationships in Blender is possible, it requires additional constraints, such as consistent ground geometry, making controlled simulation more difficult. Instead, we approximate 3D relationships using depth cues within a 2D projection, enabling scale variation and occlusion while preserving controllability. As a result, the model must reason about implicit 3D structure (e.g., front/behind) despite operating on the same 2D spatial relation labels.

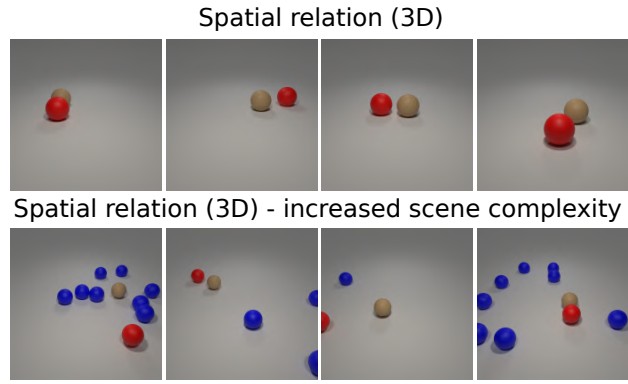

*Figure A.23.* **Examples of 3D Spatial Relations.** The setting varies camera perspective to introduce depth, scale changes, and occlusions while preserving controlled spatial relation labels.

The results for concept generalization are summarized in

Figure A.24, where we compare spatial relation performance between the 2D and 3D settings. Under the default setting with two objects, we observe comparable performance between 2D and 3D. Under increased scene complexity, with up to ten objects including distractors, we observe a slight improvement in the 3D setting compared to the 2D setting, particularly in the low-data regime (10k). We hypothesize that this may be due to additional visual cues, such as depth and scale, which provide stronger signals for learning spatial relationships compared to purely angle-based 2D representations.

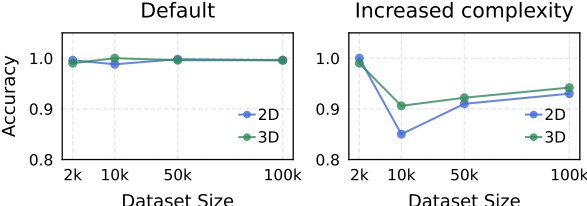

*Figure A.24.* **Spatial relation performance comparison between 2D and 3D settings.** We compare concept generalization performance under the default two-object setting and the more complex setting with distractors.

**Counting grid experiments with realistic object and background variations.** We further extend the Counting experiments with random and grid layouts by introducing (1) text-based conditioning, as described above, (2) higher-resolution images at 256×256, (3) more complex visual variations, including background textures inspired by CLEVR-Tex (Karazija et al., 2021) and more realistic objects such as cars from COMFORT (Zhang et al., 2024), and (4) larger models with approximately 200M parameters. Examples of the resulting scenes are shown in Figure A.25. As summarized in Table A.7, we observe trends analogous to the original Counting experiments: introducing a grid layout significantly stabilizes Counting, especially in the low-data regime.

*Table A.7.* **Counting accuracy under random and grid layouts in the realistic car setting.**

|  | 10k | | 100k | |
| --- | --- | --- | --- | --- |
|  | Random | Grid | Random | Grid |
| Accuracy | 0.700 | 0.992 | 0.872 | 0.998 |

random (non-grid)                                    grid

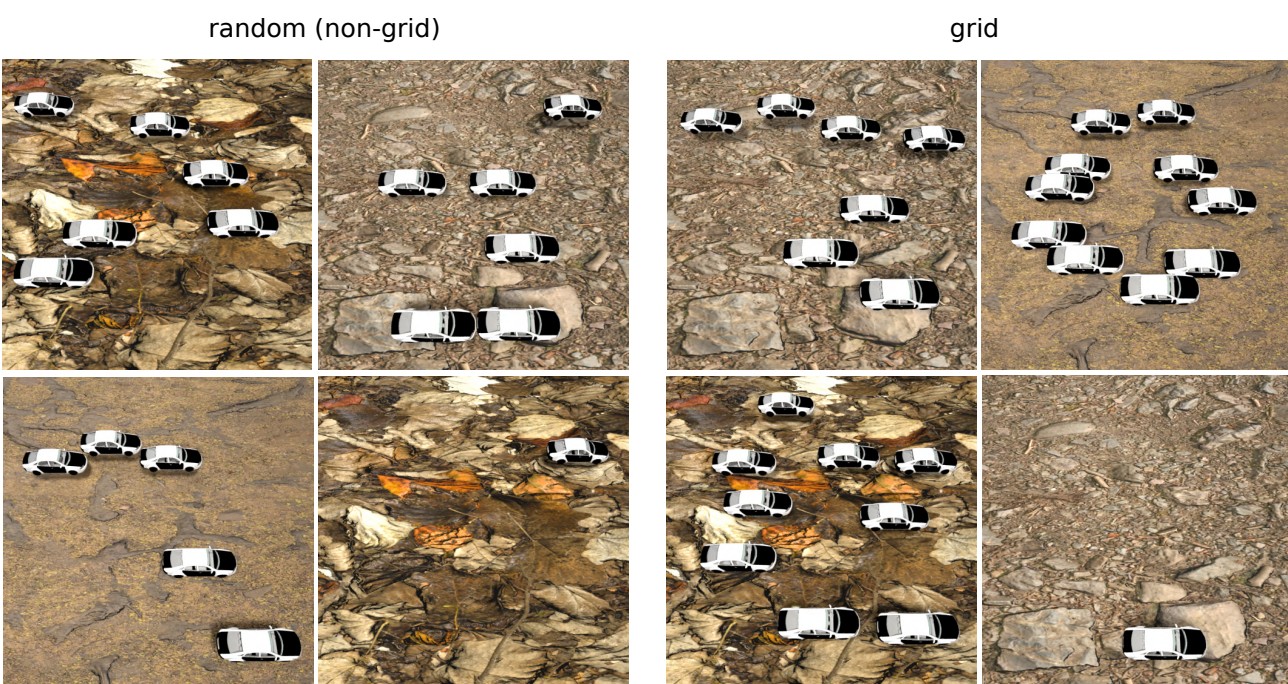

*Figure A.25.* **Examples of counting cars with diverse backgrounds.** This setting introduces more realistic object appearances and background textures.

## A.3. Qualitative Examples.

In this section, we show some qualitative examples from MOSAIC and generated samples which is trained on MOSAIC.

### A.3.1. TRAINING SAMPLES FROM MOSAIC.

Figures A.26 and A.27 show training samples from MOSAIC used in concept generalization (RQ1) and compositional generalization (RQ2), respectively, under the default setting (no increased scene complexity and no grid effect) Figure A.28 shows examples under the grid setting, used in RQ1 (Counting) and RQ2 (Counting and Spatial relations).

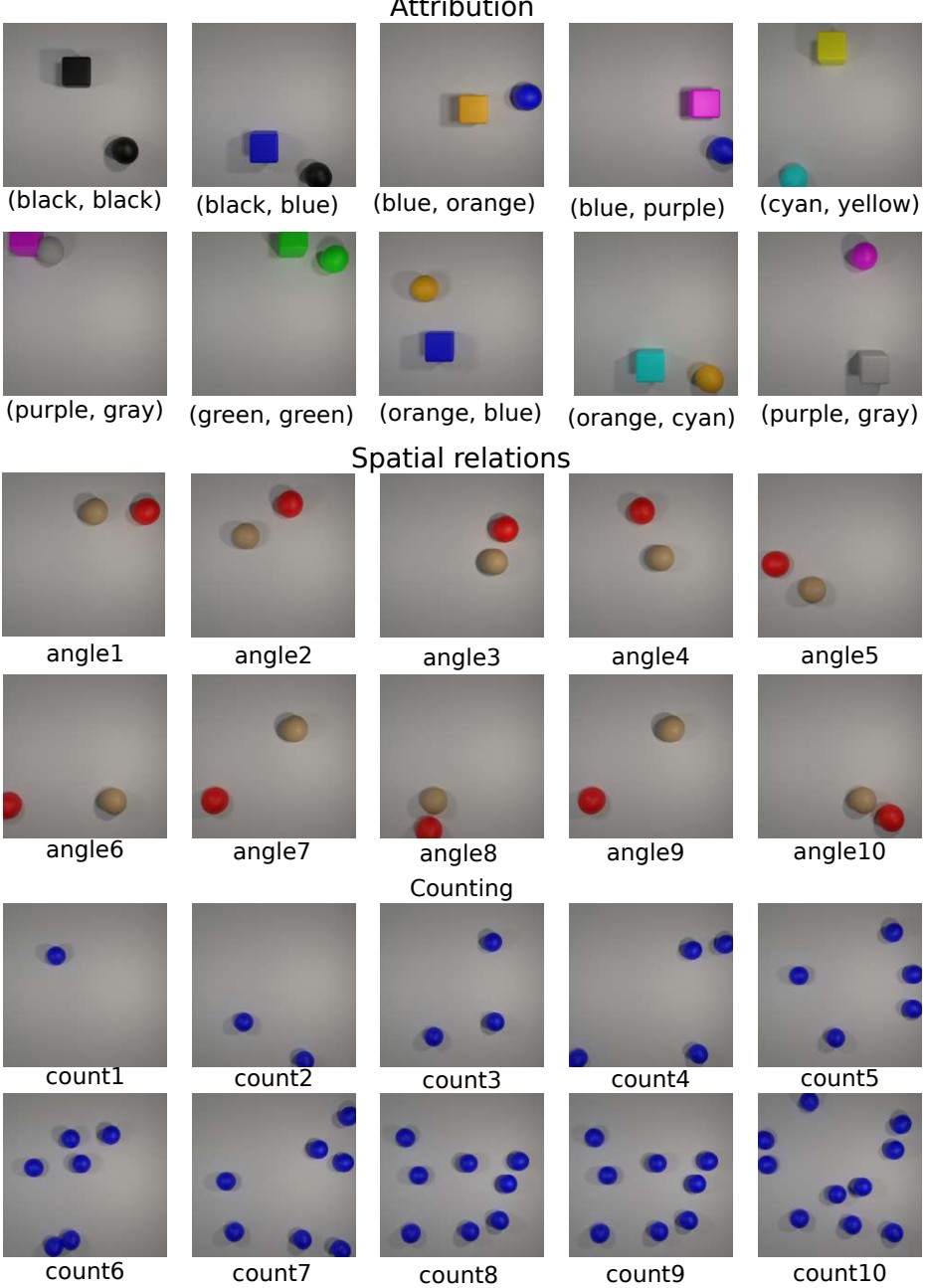

Figure A.26. **Training samples from MOSAIC (non-grid, RQ1).** Examples used for in-distribution evaluation across counting, spatial relations, and attribute-binding tasks.

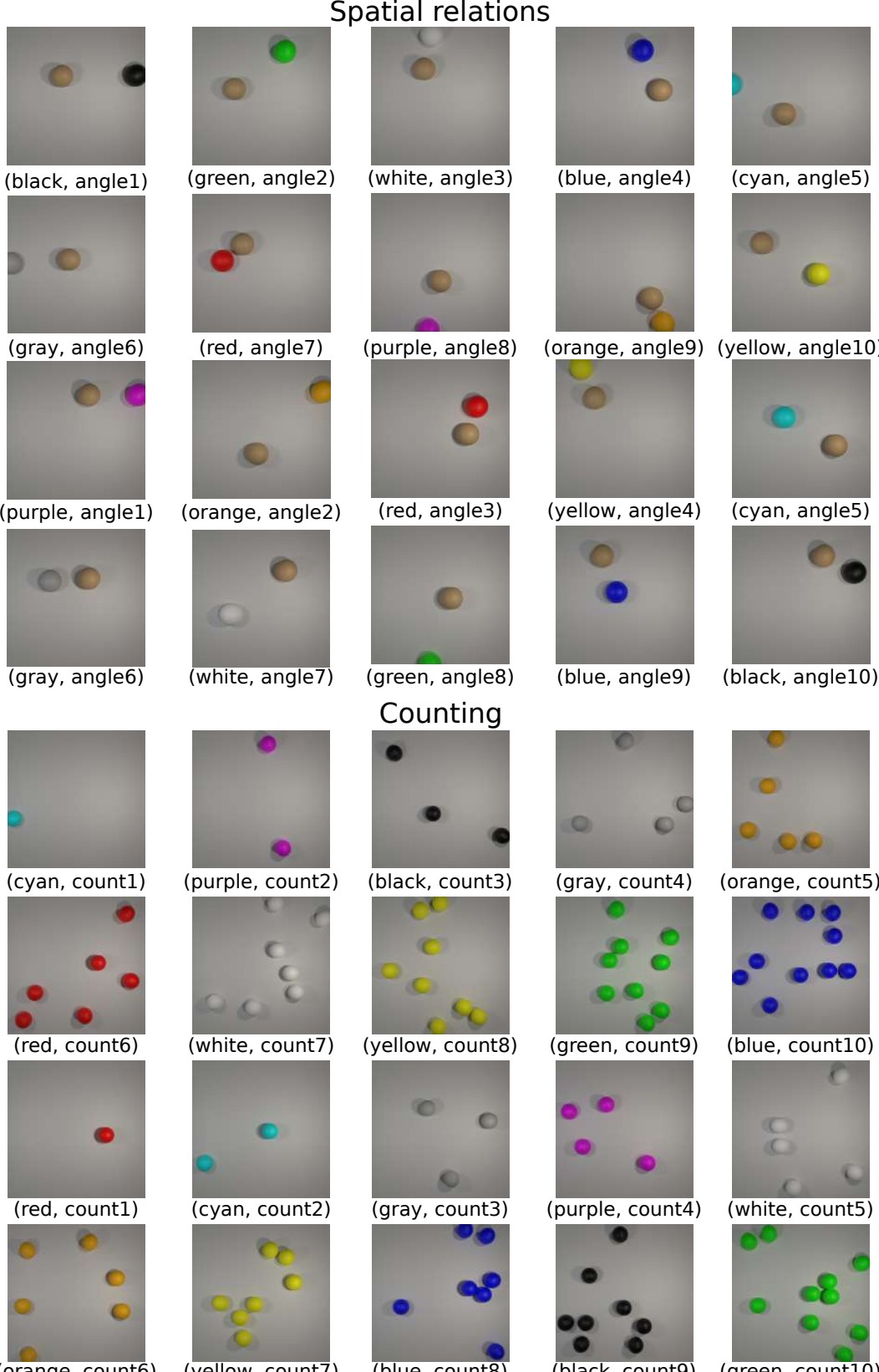

*Figure A.27.* **Training samples from MOSAIC (non-grid, RQ2).** Examples where half of compositional combinations are withheld during training to evaluate generalization to unseen compositions.

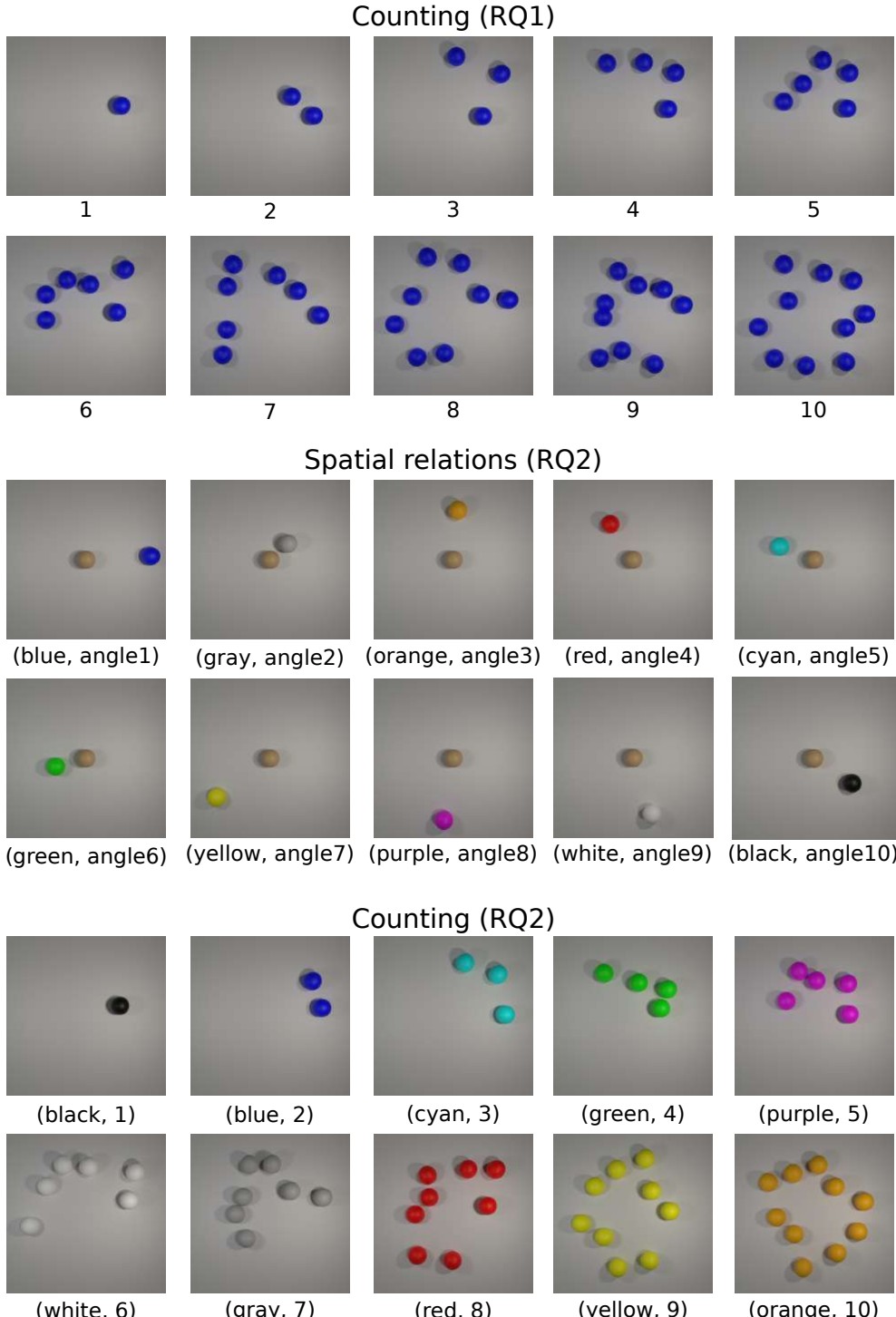

*Figure A.28.* **Training samples from MOSAIC under grid layout (RQ1 and RQ2).** Explicit spatial priors simplify scene structure, leading to improved counting and relational stability but reduced texture variation.

A.3.2. TRAINING SAMPLES FROM SPEC.

Figure A.29 shows example images from the SPEC benchmark (Peng et al., 2024), illustrating the two subsets: *Relative Spatial Relations* and *Counting*.

### Relative spatial

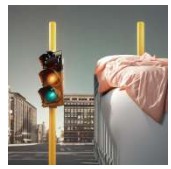 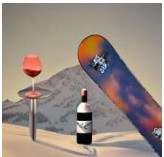 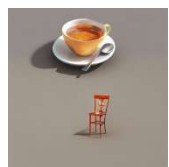 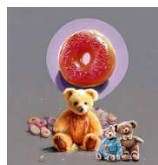 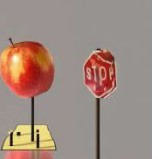

| the traffic light is positioned on the left of the bed. | the snowboard is situated to the right of the wine glass. | the cup is on top of the chair. | the donut is above the teddy bear. | the stop sign is situated to the right of the apple. |

### Counting

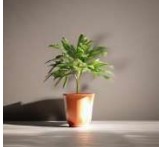 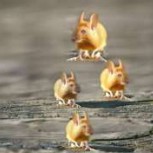 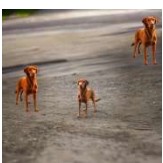 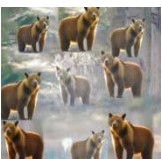 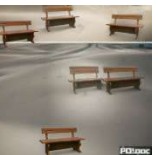

| a photograph capturing one potted plant. | an image featuring four mouse. | an image featuring three dog. | a photo displaying nine bear. | an image featuring six bench. |

*Figure A.29.* **Training samples from SPEC.** (Top) Relative Spatial Relations. (Bottom) Counting.

### A.3.3. GENERATED SAMPLES.

We present unfiltered generated samples to illustrate raw qualitative behavior, using models trained on the 100k uniform dataset. Figure A.30 shows samples for concept generalization (RQ1). To visualize how generation evolves during denoising, Figure A.31 shows intermediate outputs at different timesteps from the same noise initialization. Figure A.32 shows samples for compositional generalization (RQ2) under the setting where five diagonals are removed (unseen compositions). We omit seen examples for RQ2, as they closely resemble those in Figure A.30, differing primarily in color assignments for spatial relations and counting.

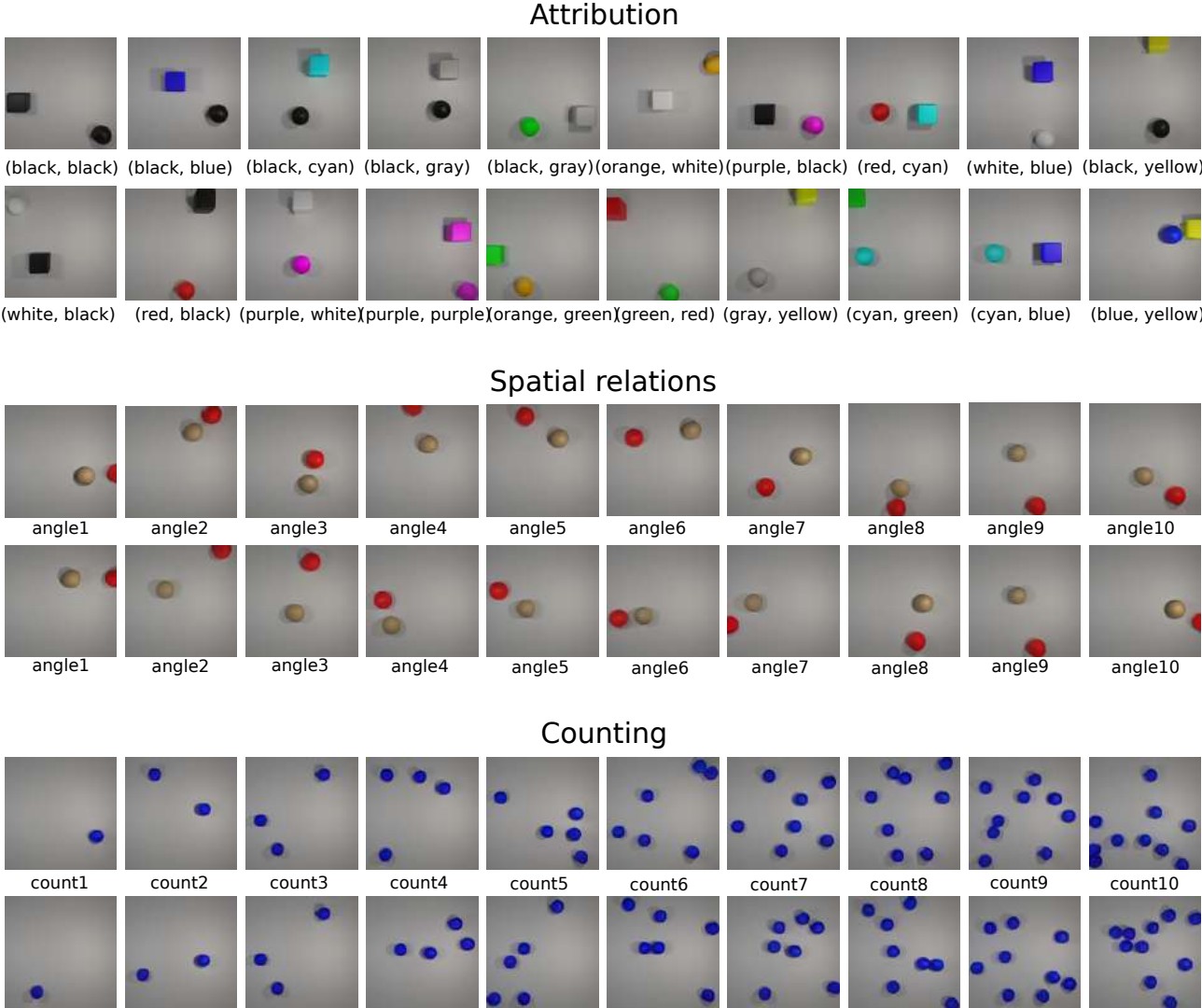

*Figure A.30.* **Generated samples for concept generalization (non-grid, RQ1).** Examples from the best-performing checkpoints trained on the 100k uniform dataset. Rows show (top) Attribution, (middle) Spatial Relations, and (bottom) Counting. Samples are shown without filtering for correctness to illustrate raw generative behavior.

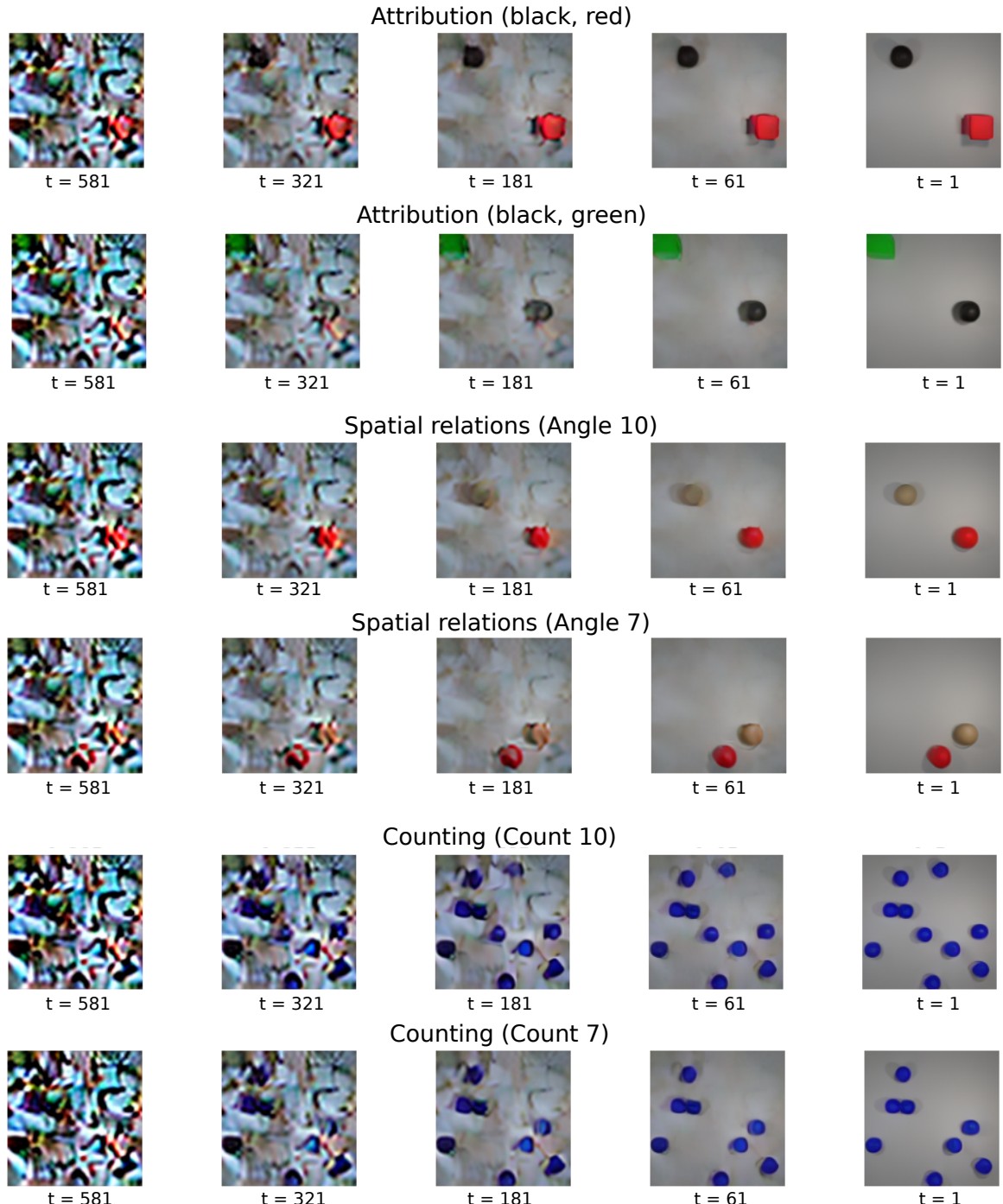

*Figure A.31.* **Generation trajectory across diffusion timesteps.** Global spatial structure is established early in the denoising process, while fine-grained shapes are refined in later steps.

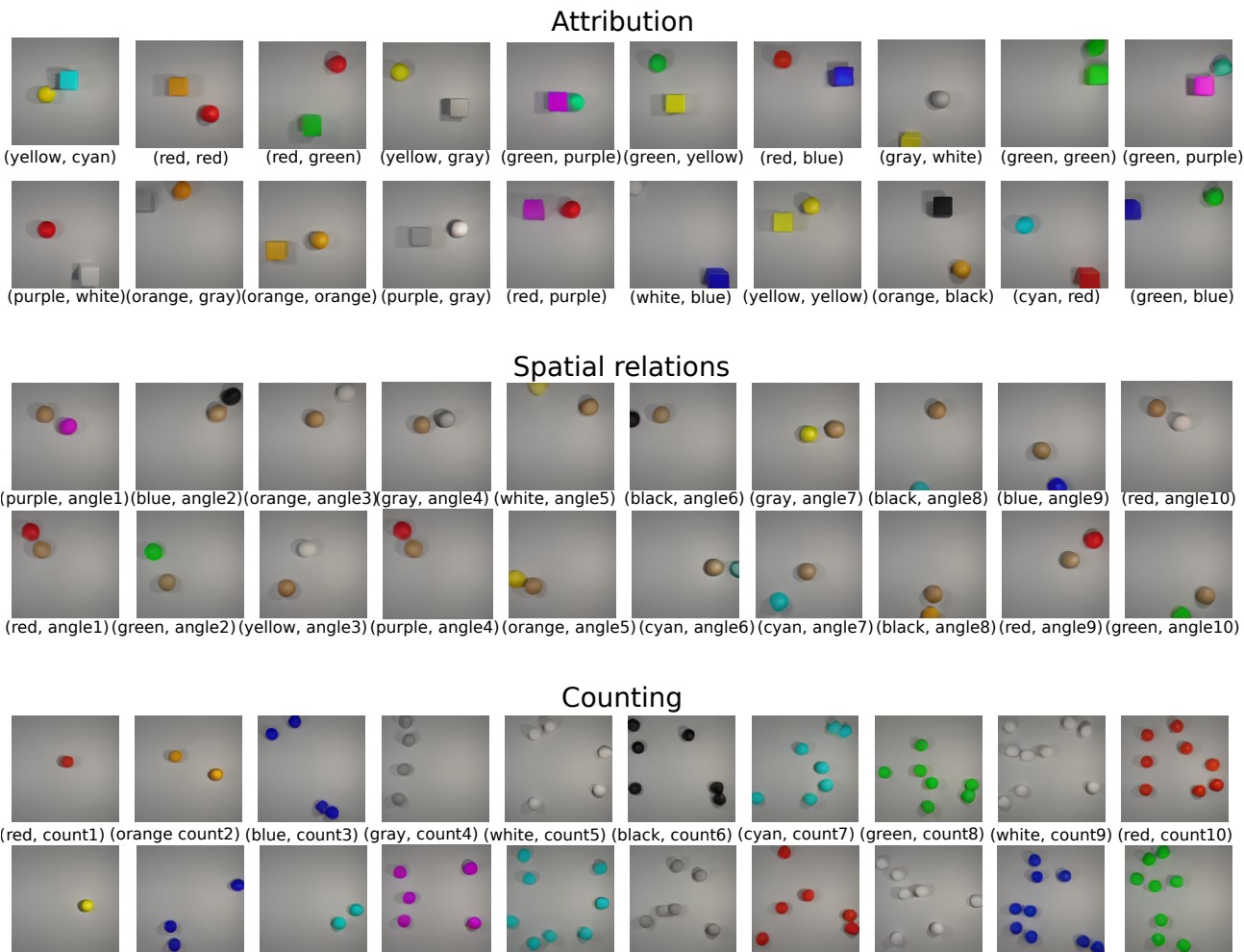

*Figure A.32.* **Generated samples for compositional generalization (non-grid, RQ2).** Results on unseen compositions when five diagonals are removed (100k dataset, best-validation checkpoint). Samples are shown without filtering for correctness to illustrate raw generation behavior under strong compositional shift.

