# OpenReview forum: "When Do Diffusion Models Learn to Generate Multiple Objects?"
_ICML.cc/2026/Conference — ICML 2026 regular_

### Official Review · Reviewer_r8Ty · 2026-03-11

**Soundness:** 2
**Presentation:** 2
**Significance:** 2
**Originality:** 2
**Overall Recommendation:** 3
**Confidence:** 4

**Summary:**

### Summary

This paper studies multi-object compositional generation in diffusion models through a controlled synthetic benchmark called MOSAIC. It argues that scene complexity is a more important source of failure than concept imbalance, and that compositional generalization degrades sharply as more combinations are withheld during training.

**Compliance With Llm Reviewing Policy:**

Affirmed.

**Final Justification:**

I appreciate the authors’ effort in clarifying the experimental setup and strengthening the empirical case in the rebuttal. The benchmark is useful, and the controlled study does provide some interesting observations about attribution, counting, and spatial relations.

However, my overall assessment remains unchanged. My main concern is that the paper still feels primarily like a benchmark/dataset paper with an empirical study, rather than a main-track paper with a strong methodological or conceptual contribution. The rebuttal improves clarity, but it does not change this core impression.

More specifically, I still do not see a sufficiently clear and coherent framework for analyzing the problem and deriving corresponding solution directions. The paper presents observations and experiments, but the overall narrative feels relatively engineering-driven and lacks the kind of structured methodological development I would expect. For example, papers such as Compositional Abilities Emerge Multiplicatively: Exploring Diffusion Models on a Synthetic Task also use a synthetic controlled setting, but they are organized around a clearer analytical framework and a more explicit explanation of the underlying phenomenon. In comparison, this paper does not give me the same level of conceptual depth or logical structure.

Overall, while I find the benchmark useful, the rebuttal only partially addresses my concerns and does not change my final judgment. I therefore maintain my original score and recommendation.

**Key Questions For Authors:**

see weakness

**Limitations:**

yes

**Strengths And Weaknesses:**

## Strengths

The paper addresses a relevant problem. Failures in multi-object generation are well known in practice, but many prior studies discuss them in broad terms and do not separate different sources of error. A useful part of this submission is that it breaks the problem into attribution, counting, and spatial relations under a controlled setup. That makes the empirical analysis easier to interpret than a single benchmark score.

The benchmark itself is also a meaningful contribution. Even though it is synthetic, it gives the authors a way to control concept frequency, object count, and scene structure more directly than in open-domain datasets. The finding that counting behaves differently from attribution and spatial relations, and that an explicit spatial prior can help in that setting, is one of the more concrete observations in the paper.

## Weaknesses

The paper's main weakness is limited originality. Its core contribution is a controlled benchmark and an empirical study, while the generative models, training objectives, and conditioning pipeline are mostly standard. As a result, the work reads more like a diagnostic benchmark paper than a method paper, and the technical novelty is not strong enough for the level of its claims.

The method side is also not especially convincing. The paper does not introduce a new learning principle or a broadly useful mechanism for improving multi-object compositional generation. The only concrete mitigation in the main text is a simple grid-based spatial prior, and its benefit is mostly shown for counting in a toy setting. This makes the paper stronger at identifying problems than at offering a meaningful way to address them.

The empirical evidence does not fully support the scope of the conclusions. Most experiments are conducted on small latent diffusion models trained on low-resolution synthetic data with one-hot conditioning, and the main evaluation depends heavily on in-domain task-specific classifiers. That setup is acceptable for a narrow controlled study, but it is too limited to support broad claims about diffusion models in general.

Some of the paper's key claims are also not isolated cleanly enough. In particular, the argument that scene complexity matters more than concept imbalance is based on interventions that change several factors at once, including object count, distractors, and layout difficulty. This weakens the causal interpretation and makes the main takeaway less secure than the paper suggests.

---

> ### Author Rebuttal · Authors · 2026-03-31
>
> We sincerely thank the reviewer for acknowledging the interpretability of failures in multi-object generation and the meaningful contribution of our benchmark. We address the concerns below.
>
> ## 1. The work reads more like a diagnostic benchmark paper than a method paper
>
> We thank the reviewer for the comment. Our goal is not to propose a new generative method, but to provide a **controlled diagnostic study** of multi-object compositional generation.
>
> - Prior work has studied compositionality either under real-world data with frequency imbalance [1], without controlled training or isolation of factors, or in simplified single-object toy settings [2].
> - In contrast, we design a fully controlled multi-object benchmark (MOSAIC) that systematically varies concept types, data distribution, and compositional splits, enabling us to directly attribute failures specifically in **multi-object cases**.
> - This reveals **non-trivial behaviors not observable in prior settings**, including:
>   (i) degradation with increasing scene complexity,
>   (ii) unstable learning in counting (e.g., collapsing), and
>   (iii) systematic compositional collapse under held-out combinations.
> - We therefore argue that the originality lies in enabling **controlled, causal analysis of compositional failures**.
>
> [1] Role Bias in Diffusion Models: Diagnosing and Mitigating through Intermediate Decomposition (CVPR 2025)
>
> [2] Compositional Abilities Emerge Multiplicatively: Exploring Diffusion Models on a Synthetic Task (NeurIPS 2023)
>
> ## 2. Lacking methodology
>
> - Our goal is not to propose a new method, but to identify when and why current diffusion models fail in multi-object compositional generation, and thereby reveal which inductive biases become important.
> - We apply the grid-based spatial prior to Counting, as it is the only task that exhibits clearly unstable learning dynamics.
> - Please refer to our detailed response to Reviewer tRq1 (Section 2: Potential solutions).
>
> ##  3. Limited experimental scale
>
> We thank the reviewer for the thoughtful feedback.
>
> - Regarding scale, controlled small-scale setups are standard in prior work [1,2,3], many of which rely on even simpler settings (e.g., lower resolutions at 32x32~80x80 pixels, 2D single-object scenes).
> - In contrast, our setting is more realistic than prior controlled studies: we consider multi-object scenes rendered by simulation. Models are trained at resolution 128.
> - To further address the reviewer's concern, we extend our experiments on Counting with/without grid settings by introducing
>     - (1) text-based conditioning (Details are in the rebuttal to Reviewer tRq1 1. Simplified condition encoding),
>     - (2) 256 resolution,
>     - (3) more complex visual variations (background texture) and more realistic objects (like cars),
>     - (4) scaling the size of models (200M params).
>     - Examples of the scenes are presented in [Figure 9](https://bit.ly/4c2sJ1n).
> - We observe analogous results to Fig. 9 in the main paper: introducing a grid significantly stabilizes Counting, especially for the low-data regime.
>     | Counting | 10k (Random) | 10k (Grid) | 100k (Random) | 100k (Grid) |
>     | :----| :---- | :---- | :---- | :---- |
>     | Accuracy|0.7|0.992|0.872|0.998|
>
> [1] Compositional Abilities Emerge Multiplicatively: Exploring Diffusion Models on a Synthetic Task (NeurIPS 2023)
>
> [2] Why Diffusion Models Don’t Memorize: The Role of Implicit Dynamical Regularization in Training (NeurIPS 2025)
>
> [3] Emergence of Hidden Capabilities: Exploring Learning Dynamics in Concept Space (NeurIPS 2024)
>
> ## 4. Causal Interpretation
>
> We thank the reviewer for raising this important point. We would like to clarify that the factors discussed correspond to **two separate, controlled analyses**, and are not confounded in our experiments.
>
> *1. Scene Complexity Factors*
>
> - In Figure 7, the layout is not varied. The only factors considered are:
>   (i) number of target objects, and (ii) number of distractors.
> - Our goal is to compare tasks under **matched scene complexity**. Since Counting naturally involves multiple objects (up to 10), we increase complexity in other tasks as follows:
>   - (i) Attribution: increasing the number of target objects as Counting
>   - (ii) Spatial Relations: increasing the number of distractors
> - We note that increasing target objects in spatial relations would make the relation ill-defined, while adding distractors preserves the task semantics (relations are typically defined between a subset of objects).
>
> *2. Role of Layout (Separately Controlled)*
>
> - Layout is studied separately in main Figure 9, where we isolate a single factor: **unstructured vs. structured (grid) layouts**.
> - Therefore, layout is **cleanly isolated as an independent factor**, and the improvement can be attributed to reduced spatial complexity rather than other confounding variables.
>
> Overall, our experimental design explicitly separates scene complexity and layout, supporting our causal interpretation.

---

> > ### Author Rebuttal · Reviewer_r8Ty · 2026-04-02
> >
> > I have read the rebuttal carefully and appreciate the clarifications. The response helps strengthen the empirical case for the benchmark and makes some of the experimental conclusions clearer.
> >
> > However, my main concern remains. In my view, the core contribution of the paper is still the dataset/benchmark and the controlled empirical study built around it. While this is a useful contribution, it seems better aligned with a dataset or benchmark track than with the standard I would expect for a main-track paper. My reservation is mainly about the level of theoretical or methodological contribution: the paper does not yet provide a sufficiently strong general principle, conceptual abstraction, or broadly useful mechanism beyond the benchmark itself.
> >
> > For this reason, I consider my concerns only partially resolved, and not in a way that can be fully addressed within a short rebuttal.

---

> > > ### Author Response · Authors · 2026-04-02
> > >
> > > We thank the reviewer for carefully reading our rebuttal and for acknowledging that it strengthens the empirical case and clarifies the experimental conclusions. We also note that the reviewer indicated follow-up questions, but none were posed.
> > >
> > > Regarding the remaining concern about positioning: we observe that the reviewer's technical concerns from the initial review (causal interpretation, experimental scale, conditioning mechanism) have been substantively addressed in our rebuttal, and the remaining disagreement centers on whether a diagnostic benchmark study constitutes a main-track contribution.
> > >
> > > We respectfully note that ICML does not restrict benchmark- or evaluation-driven studies to a separate track, and such works have consistently appeared in the main track when they provide meaningful insights into model behavior. Representative examples of (i) empirical and diagnostic studies and/or (ii) benchmark-oriented works in the main track, across ICML and other top-tier venues, include [1,2,3,4,5,6,7].  Our work similarly goes beyond proposing a dataset: it reveals non-trivial failure modes (counting collapse, compositional degradation scaling) **EDIT (07. April)**: that were not observable in prior settings by systematically disentangling key factors. Moreover, we provide concrete experimental evidence that characterizes when generalization succeeds or fails, offering actionable guidance for future method development.
> > >
> > > We believe that, given the absence of unresolved technical concerns, this constitutes a sufficient contribution for the main track.
> > >
> > > **EDIT (07. April)**: We acknowledge that [2] makes an important contribution toward understanding compositional behavior in a single-object regime under continuous-condition settings. Their work provides valuable empirical insights in a controlled setting. Similarly, our work also adopts a controlled setup; we study **multi-object** settings and the dependence on data scale under discrete-condition settings, which more closely reflect many real-world compositional scenarios (e.g., attribution, spatial relations, and counting). We further disentangle multiple key factors systemically, including composition coverage, data imbalance, and scene complexity. This allows us to organize our analysis in a structured and causal manner, providing insights into when and why generalization succeeds or fails in diffusion models.
> > >
> > > [1] How Far is Video Generation from World Model: A Physical Law Perspective (ICML 2025)
> > >
> > > [2] Compositional Abilities Emerge Multiplicatively: Exploring Diffusion Models on a Synthetic Task (NeurIPS main track 2025)
> > >
> > > [3] When and How Does CLIP Enable Domain and Compositional Generalization? (ICML 2025)
> > >
> > > [4] LaRA: Benchmarking Retrieval-Augmented Generation and Long-Context LLMs – No Silver Bullet for LC or RAG Routing (ICML 2025)
> > >
> > > [5] Exploring the Effectiveness of Object-Centric Representations in Visual Question Answering: Comparative Insights with Foundation Models (ICLR 2025)
> > >
> > > [6] EMBODIEDBENCH: Comprehensive Benchmarking Multi-modal Large
> > > Language Models for Vision-Driven Embodied Agents (ICML 2025)
> > >
> > > [7] In search of Forgotten Domain Generalization (ICLR 2025)

---

### Official Review · Reviewer_hJXJ · 2026-03-11

**Soundness:** 3
**Presentation:** 4
**Significance:** 3
**Originality:** 3
**Overall Recommendation:** 5
**Confidence:** 4

**Summary:**

The paper introduces mosaic, a controlled framework that isolates why diffusion models struggle with multi-object attribution, counting, and spatial relations. It reveals a unique 'counting collapse' where models memorize at small data scales but fail at intermediate levels before finally generalizing at large scales. The research finds that scene complexity drives failures more than data imbalance and demonstrates that spatial inductive biases, such as grid layouts, significantly improve reliability. Ultimately, the study establishes a difficulty hierarchy, "Attribution < Counting < Spatial Relations", showing that spatial reasoning is the most challenging task for compositional generalization.

**Compliance With Llm Reviewing Policy:**

Affirmed.

**Final Justification:**

Thanks for the detailed rebuttal response from the authors. I maintain my initial positive score.

**Key Questions For Authors:**

* While the mosaic dataset focuses on the repeated generation of identical objects, I am curious whether the same collapse phenomenon occurs when repeating a variety of different, heterogeneous objects.
* The paper defines spatial relationships solely through 2D angles. While this simplification is understandable, real-world images frequently involve 3D spatial arrangements, raising questions about how these results might translate to 3D spatial contexts.

**Limitations:**

yes

**Strengths And Weaknesses:**

**Strength**
- The paper is well-written and easy to follow
- Beyond mere observation, the study achieves complete variable isolation through a strictly controlled environment called 'mosaic'.
- It utilizes both UNet and DiT architectures and provides an analysis of the impact of model parameter scaling.
- By adding finetuning experiments on a real-world model, the study proves that findings from synthetic data are valid in practical environments.
- While previous studies often focus on reporting phenomena, this work is impressive for investing root causes by actively manipulating variables.

**Weakness**
- While the study successfully clarifies "when" the collapse occurs, the in-depth architectural or mathematical analysis of why the unique U-shaped curve appears specifically in counting tasks seems somewhat lacking compared to the experimental observations.
- In most experiments, the number of training steps appears to be fixed at 20,000. When the dataset size is 2k, 20k steps means each image is seen 10 times, whereas for a 100k dataset, less than 20% of the data is even processed. This makes it ambiguous whether the observed transition between memorization and generalization is driven purely by data scale or if it is an artifact of over-training on smaller datasets.
- Most core conclusions are derived from the "mosaic" dataset, which is highly simplified (geometric shapes on a gray background). While the use of a controlled environment is commendable, the fine-tuning experiments on Stable Diffusion 3 alone may not be sufficient to demonstrate that these mechanisms remain robust in more complex, real-world data environments.

---

> ### Author Rebuttal · Authors · 2026-03-31
>
> We appreciate that the reviewer acknowledges complete isolation in a strictly controlled environment, the validity of findings from synthetic data in practical environments, and the importance of investing in root causes by actively manipulating variables.
>
> ## W1. The in-depth architectural or mathematical analysis
>
> We thank the reviewer for this important point.
>
> - We agree that a deeper analysis of why the U-shaped (collapse) behavior occurs in counting is important.
> - However, we emphasize that identifying when and under what conditions such failures arise is a necessary first step.
> - We analyze this behavior through per-class dynamics (Appendix Figures 19, 20, and 21)
> - It shows that larger counts collapse early, are difficult to learn, and are only memorized later during training, resulting in the observed U-shaped curve.
> - While a full theoretical explanation is beyond the scope of this work, our controlled setup provides a foundation for future investigation into the underlying causes.
>
> ## W2. Fixed steps
>
> We thank the reviewer for this important observation. We give an answer based on our understanding of the question; if we have misunderstood any aspect of it, we would appreciate further clarification.
>
> - We clarify that all results are reported using the best checkpoint selected based on validation accuracy, rather than the final training step. This reduces the risk of overtraining artifacts in smaller datasets. We further analyze memorization behavior in Appendix Figures 16 and 17.
> - We use a fixed number of gradient optimization steps across datasets to study training dynamics [1]. To account for differences in data exposure, we further analyze performance across training epochs for different dataset sizes (2k and 10k) ([Figure 3](https://bit.ly/4c2sJ1n)).
>
> [1] Why Diffusion Models Don’t Memorize: The Role of Implicit Dynamical Regularization in Training (NeurIPS 2025)
>
> ## W3. Simplicity of MOSAIC
>
> - While our setting is still controlled, we have taken concrete steps to bridge the gap toward more realistic scenarios in our response to Reviewer r8Ty (3. Limited experimental scale).
> - We extend MOSAIC beyond simple geometric shapes and introduce diverse textured backgrounds.
> - In particular, we incorporate real-world objects (e.g., cars) and vary background textures, resulting in scenes with greater visual diversity.
> - Despite these changes, we observe consistent trends with our finding of concept generalization, suggesting that the identified failure modes are not limited to simple settings.
>
> ## Q1. Counting with heterogeneous objects
>
> We thank the reviewer for this insightful suggestion.
>
> - Following the reviewer’s suggestion, we construct an additional setting where object attributes (shape, size, and color) are randomly varied, resulting in heterogeneous object compositions (examples shown in [Fig. 4](https://bit.ly/4c2sJ1n).
> - We evaluate counting performance under this setting and compare it with the original setup using identical objects. The results for concept generalization are summarized in [Fig. 5](https://bit.ly/4c2sJ1n).
> - These results suggest that counting depends on consistent object representations, which degrade as object diversity increases, leading to more severe failures.
> - We also observe that the collapse persists under this heterogeneous object setting, as shown in [Fig. 6](https://bit.ly/4c2sJ1n).
> - We will discuss these in the revision.
>
> ## Q2. 3D Spatial relationship performance
>
> Thank you for a great question.
>
> - To investigate this, we introduce a more realistic geometric setting by varying camera perspectives, which induces depth, scale changes, and occlusions ([Fig. 7](https://bit.ly/4c2sJ1n)).
> - While constructing 3D relationships with Blender is possible, it requires additional constraints, such as consistent ground geometry, making controlled simulation difficult.
> - Instead, we approximate 3D relationships using depth cues within a 2D projection, enabling scale variation and occlusion while preserving controllability.
> - As a result, the model must reason about implicit 3D structure (e.g., front/behind) despite operating on the same 2D spatial relations.
> - The results of concept generalization are summarized in [Fig. 8](https://bit.ly/4c2sJ1n) comparing spatial relation performance between 2D and 3D settings on concept generalization settings.
> - Under the default setting (2 objects), we observe comparable performance between 2D and 3D.
> - Under increased scene complexity (up to 10 objects with distractors), we observe a slight improvement in the 3D setting compared to the 2D setting, particularly in low-data regimes (10k).
> - We hypothesize that this may be due to additional visual cues (e.g., depth and scale) in 3D scenes, which could provide stronger signals for learning spatial relationships compared to purely angle-based 2D representations.
> - We will include these results and the discussion in the revision.

---

> > ### Author Rebuttal · Reviewer_hJXJ · 2026-04-04
> >
> > I would like to thank the authors for their thorough and thoughtful rebuttal. I especially appreciate how my concern regarding the epoch imbalance issue (W2) was neatly resolved through the use of validation checkpoints and the additional epoch-based analysis. I am satisfied with the response and will maintain my positive score.

---

> > > ### Author Response · Authors · 2026-04-05
> > >
> > > We are glad the rebuttal addressed your concerns. We truly appreciate the reviewer's feedback and acknowledgment, and we will include the rebuttal discussion in the revision.

---

### Official Review · Reviewer_tRq1 · 2026-03-13

**Soundness:** 3
**Presentation:** 3
**Significance:** 3
**Originality:** 3
**Overall Recommendation:** 4
**Confidence:** 3

**Summary:**

This paper investigates why diffusion models struggle with multi-object generation and introduces MOSAIC, a controlled benchmark for analyzing three core capabilities: attribute binding, counting, and spatial relations. The paper shows that scene complexity is a more critical factor than concept imbalance, that counting is particularly fragile in low-data regimes, and that compositional generalization deteriorates sharply as more concept combinations are excluded from training, with spatial relations being the most challenging case.

**Compliance With Llm Reviewing Policy:**

Affirmed.

**Final Justification:**

The authors have partially addressed my concerns. Therefore, I decided to maintain my current score.

**Key Questions For Authors:**

If the authors can address these concerns, I would be willing to raise my score during the rebuttal stage.

**Limitations:**

1. The conditioning mechanism is overly simple.

2. The paper stops at empirical diagnosis and does not further progress toward potential solutions.

**Strengths And Weaknesses:**

Strengths:

1. The paper is very well written, with a clear motivation, methodology, and set of conclusions.

2. The experimental study is thorough and comprehensive.

3. A key strength of this work is that it disentangles three types of compositional abilities—attribute binding, counting, and spatial relations—and studies them separately. The findings suggest that failures in multi-object generation cannot be attributed solely to long-tail frequency imbalance; rather, scene complexity and missing concept compositions play a more important role. In particular, counting is especially vulnerable in low-data settings, while spatial relations exhibit the weakest compositional generalization, highlighting the lack of strong compositional inductive biases in current diffusion models.

Weaknesses:

1. The conditioning mechanism used in this work is rather simple (i.e., one-hot condition injection). This design choice may prevent the diffusion models from fully realizing their representational capacity, which could in turn affect the strength of the paper’s conclusions.

2. Given these findings, it would significantly strengthen the paper if the authors could further explore whether some simple yet effective strategies can be designed to improve performance on attribute binding, spatial relations, and counting. I understand that this paper is primarily diagnostic in nature, but even a preliminary attempt at addressing the identified issues would substantially increase its practical value.

---

> ### Author Rebuttal · Authors · 2026-03-31
>
> We are grateful for the reviewer’s positive feedback, particularly for recognizing that we disentangle three types of compositional abilities and for the paper's well-written, thorough presentation. We address the reviewer’s concerns and questions below.
>
> ## 1. Alternative conditioning mechanism
>
> We thank the reviewer for raising this important point.
>
> - Our original design choice of using one-hot condition encoding was intended to isolate the effect of compositional generalization, without introducing additional compositional capacity from the encoder itself. To this end, we jointly train the condition encoder with the diffusion model.
> - We agree, however, that such a simplified encoding may not fully capture all real-world settings.
> - To address this, we conduct additional experiments using a CLIP-B/16 text encoder as the conditioning module for a DiT backbone, with the encoder kept frozen during training.
> - We design prompts aligned with MOSAIC factors.
>     - Attribution: “A photo of a black sphere and a red cube”,
>     - Counting: “A photo of ten blue spheres”, and
>     - Spatial relations: “A photo of the red sphere is at an angle of 222° from a brown sphere, measured from the rightward direction” (in practice, we use angle ranges such as 215°–234°).
> - The results ([Figure 1](https://anonymous.4open.science/api/repo/icml26-6632/file/rebuttal.pdf?v=fb328f47)) show a slight improvement in concept generalization settings at 10k dataset size, when using CLIP text embedding as conditioning, compared to the baseline.
> - We also run an experiment over compositional generalization ([Figure 2](https://anonymous.4open.science/api/repo/icml26-6632/file/rebuttal.pdf?v=fb328f47)): the gains there are more noticeable.
> - Therefore, to further isolate the effect of the encoder for the compositional generalization, we include an additional baseline with a frozen condition encoder trained separately using one-hot inputs with BCE loss (denoted as “Baseline (frozen)” in the figure), corresponding to the experiments in Appendix Figure 29 (UNet). We find that its performance is comparable to CLIP-based conditioning.
> - Importantly, the overall trends remain unchanged: (i) counting remains difficult in low-data regimes, and (ii) performance consistently degrades as more compositions are held out.
> - These results suggest that the observed limitations are not primarily due to the simplicity of the condition encoding, but are intrinsic to the compositional learning challenge.
> - We note that more diverse text conditioning is also used in SD-3 experiments (Figure 12 in the main paper), further supporting that our findings are not specific to simplified prompts.
> - We will include these additional results and discussion in the revision.
>
> ## 2. Potential solutions
>
> We thank the reviewer for this important point.
>
> - We agree that developing effective mitigation strategies would increase the practical impact of this work. However, our goal is not to propose a specific solution, but to identify *when such solutions become necessary*.
> - Our findings show that diffusion models do not uniformly fail across settings. Instead, failures arise under specific conditions, most notably in low-data regimes and with increasing scene complexity. This suggests that mitigation is not always required, but becomes critical in specific regimes.
> - In this context, the grid-based prior is understood as a **diagnostic intervention**, rather than a complete solution.
> - The results (Figure 9 in the main paper) show that introducing structure and reducing scene complexity can partially stabilize training for Counting, revealing when inductive bias becomes necessary.
> - Importantly, this intervention does not consistently improve all tasks. For example, in compositional generalization for spatial relations (Appendix Figures 24 and 28), we observe trade-offs, where improvements come at the cost of degradation in other compositional factors (e.g., color), indicating that simple structural priors are insufficient as a general intervention.
> - Therefore, identifying these failure regimes and characterizing the role of inductive bias is itself a valuable contribution, as it provides concrete guidance for future method development.
> - We will clarify this positioning more explicitly in the revision.

---

> > ### Author Rebuttal · Reviewer_tRq1 · 2026-04-03
> >
> > Thanks to the author for the rebuttal.
> >
> > The author successfully addressed my concerns about the conditioning mechanism and some concerns about the potential solutions. However, in the end, I did not see a reasonable and effective solution. Therefore, I decided to maintain my positive score.

---

> > > ### Author Response · Authors · 2026-04-05
> > >
> > > We appreciate the reviewer’s feedback and acknowledgment of our rebuttal. We hope that our diagnostic results and Mosaic toolkit will enable future research and methodological development.
> > >
> > > Our findings on when compositional generation succeeds in multi-object settings show that different success and failure modes arise from distinct underlying factors (e.g., data size, scene complexity, and exposure to diverse compositions). Importantly, this diagnostic-first approach is consistent with a growing body of recent work that focuses on identifying the conditions under which models succeed or fail. Representative examples include [1-5].
> > >
> > > We thank the reviewer again for the constructive feedback!
> > >
> > > [1] How Far is Video Generation from World Model: A Physical Law Perspective (ICML 2025)
> > >
> > > [2] When and How Does CLIP Enable Domain and Compositional Generalization? (ICML 2025)
> > >
> > > [3] Exploring the Effectiveness of Object-Centric Representations in Visual Question Answering: Comparative Insights with Foundation Models (ICLR 2025)
> > >
> > > [4] Monitoring Latent World States in Language Models with Propositional Probes (ICLR 2025)
> > >
> > > [5] Two Effects, One Trigger: On the Modality Gap, Object Bias, and Information Imbalance in Contstrastive Vision-Language Models (ICLR 2025)

---

### Official Review · Reviewer_fx97 · 2026-03-13

**Soundness:** 3
**Presentation:** 3
**Significance:** 3
**Originality:** 3
**Overall Recommendation:** 5
**Confidence:** 4

**Summary:**

The authors conduct a systematic study of effect of dataset properties on multi-object generation. They specifically have conducted controlled research on the effect of dataset concept imbalance in model concept generalization and heldout composition combination in dataset in model composition generalization.To facilitate the study the authors have created a dataset generation framework which systematically varies a variable (color, count, ...) while keeping the rest of variables fixed. With the help of this generator, the authors have completed a thorough and systematic investigation on the effect of dataset in generation capabilities of the model.

**Compliance With Llm Reviewing Policy:**

Affirmed.

**Key Questions For Authors:**

No Question

**Limitations:**

yes

**Strengths And Weaknesses:**

## Strengths
- The paper is structured in a clear way, the research questions are clearly indicated and systematic approach to answer them is outlined
- The dataset generator is built systematically to be able to test different dataset properties in isolation and in a controlled fashion
- The authors also trained models to be able to quantitatively measure accuracy of the model at each training regime
- The conducted experiments are clear and follow a strategic choice
- The findings in the paper are fundamental to understand the current gaps of the model which can help in further advancing the research
## Weaknesses
- It would have been nice to explain the data generation process more, specifically whether the images are generated via generative models or simulation / programatic ways and if the first option, how the authors have verified the dataset precision.

---

> ### Author Rebuttal · Authors · 2026-03-31
>
> We sincerely thank the reviewer for the positive feedback and for recognizing the clear motivation and structured design of our study, as well as the clarity of our experimental setup. We also appreciate the thoughtful suggestions and questions, which we address below.
>
> **1. How have the authors verified the dataset precision?**
>
> - Our dataset is generated using a simulation-based pipeline, not generative models. Specifically, we build on the COMFORT [1], which uses Blender to render photorealistic scenes with explicitly controlled parameters.
> - Unlike generative approaches, all scene factors in our setup are programmatically specified before rendering. This allows us to precisely control and verify attribution, spatial relations, and object counts.
> - We extend COMFORT by explicitly parameterizing additional factors for attribution, spatial relations, and counting, using objects from COMFORT and MULTIMODAL3DIDENT [2].
> - Because the scene configuration is deterministically defined prior to rendering, the resulting annotations are exact by construction, ensuring full dataset precision without the need for post-hoc verification.
> - We will revise the manuscript to make the data generation process more explicit (currently described in Appendix A.1.2).
>
> [1] Do Vision-Language Models Represent Space and How? Evaluating Spatial Frame of Reference Under Ambiguities (ICLR 2025)
>
> [2] Identifiability Results for Multimodal Contrastive Learning (ICLR 2023)

---

> > ### Author Rebuttal · Reviewer_fx97 · 2026-04-06
> >
> > Thank you so much for taking time to address my concern. The fact that the dataset is generated programmatically makes all the paper results reliable.

---

> > > ### Author Response · Authors · 2026-04-06
> > >
> > > We sincerely appreciate the reviewer’s feedback and acknowledgment of our rebuttal, and we will update the manuscript to reflect this discussion on the dataset’s reliability.

---

### Decision · Program_Chairs · 2026-04-30

**Decision:**

Accept (regular)

**Comment:**

The submission makes a clear empirical contribution: it introduces a controlled benchmark for multi-object diffusion generation and studies failures in attribution, counting, and spatial relations.

The main weakness is that the contribution is primarily diagnostic rather than methodological, so the scope of the claims should remain aligned with the controlled setting. At the same time, the rebuttal addressed most of the concrete technical concerns raised by reviewers, including dataset precision, conditioning design, training/checkpoint selection, and added experiments in more realistic settings. Overall reviewer sentiment remained positive: three reviewers maintained accept-level recommendations, while one reviewer continued to question the paper’s conceptual novelty and framing.

Overall, this is a solid controlled study of an important failure mode in diffusion models. It does not propose a new generative method, but it offers useful empirical insight and a benchmark that the community can build on. I support **accept**.